# The Optimal Token Baseline: Variance Reduction for Long-Horizon LLM-RL

Yingru Li [*]   Jiawei Xu [* 1]   Ziniu Li [* 1]   Jiacai Liu [2]   Wei Liu [3]   Yuxuan Tong [4]   Longtao Zheng [5]   Zhenghai Xue [5]
Yaxiang Zhang [6]   Tianle Cai [4]   Ge Zhang [4]   Qian Liu   Baoxiang Wang [1 7]

## Abstract

Reinforcement Learning (RL) for Large Language Models (LLMs) often suffers from training collapse in long-horizon tasks due to exploding gradient variance. To mitigate this, a baseline is commonly introduced for advantage computation; however, traditional value models remain difficult to optimize, and standard group-based baselines overlook sequence heterogeneity. Although classic optimal baseline theory can achieve global variance reduction, it neglects token heterogeneity and requires prohibitive gradient-based computation. In this work, we derive the Optimal Token Baseline (OTB) from first principles, proving that gradient updates should be weighted inversely to their cumulative gradient norm. To ensure efficiency, we propose the Logit-Gradient Proxy that approximates the gradient norm using only forward-pass probabilities. Our method achieves training stability and matches the performance of large group sizes ($N = 32$) with only $N = 4$, reducing token consumption by over 65% across single-turn and tool-integrated reasoning tasks.

## 1 Introduction

Reinforcement Learning (RL) has become the standard paradigm for aligning Large Language Models (LLMs) in complex reasoning (Guo et al., 2025; Zeng et al., 2025) and agentic tasks (Jin et al., 2025; Xue et al., 2025). However, as tasks grow in complexity and generation horizons expand, practitioners frequently observe **Training Collapse** in RL training, where the gradient norm suddenly surges, causing model performance to crater. As shown in Figure 1, models often learn effectively for hundreds of steps before expe-

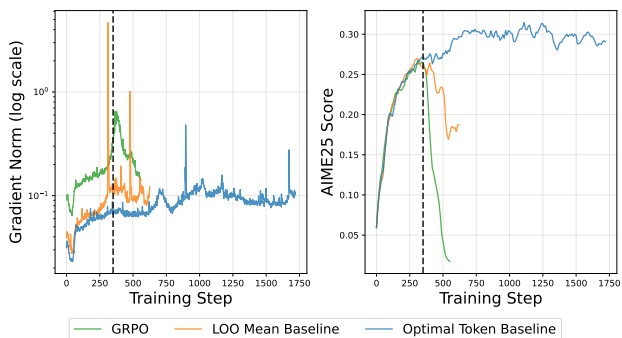

*Figure 1.* Gradient Norm and AIME25 Score under Single-Turn Reasoning. We adopt full on-policy training on the Qwen3-8B-Base. The vertical dotted line indicates the point at which the compared methods collapse, coinciding with a sudden surge in gradient norm. Notably, our Optimal Token Baseline yields a stable gradient norm, resulting in stable training and a higher score.

riencing an unexpected gradient spike, leading to a severe degradation in scores.

A key reason for such gradient norm surges is the variance in gradient estimation during policy update (Weaver & Tao, 2013). To reduce gradient variance, introducing a baseline for advantage computation is a common strategy. While actor-critic methods employ a separate value model to estimate this baseline, training such an auxiliary model is notoriously difficult in LLM-RL (Li et al., 2023). Therefore, recent methods have favored simpler REINFORCE-style baselines. For instance, ReMax (Li et al., 2023) uses the greedy response as its baseline, while GRPO (Shao et al., 2024) and RLOO (Ahmadian et al., 2024) calculate the baseline from the average reward of multiple sequences. However, these methods assume every sequence in a group is equally informative and overlook the **sequence heterogeneity**. Prior work (Dayan, 1991; Greensmith et al., 2004; Weaver & Tao, 2013) derives the Optimal Global Baseline which weights reward expectations by the total gradient magnitude of each sequence. However, it requires prohibitive gradient-based computations and critically neglects **token heterogeneity**: tokens at the same generation step exhibit varying gradient magnitudes across different sequences. It is also **acausal**, as the total gradient magnitude cannot be known during intermediate token generation steps.

[1]The Chinese University of Hong Kong, Shenzhen [2]Fudan University [3]Hong Kong University of Science and Technology [4]ByteDance [5]Nanyang Technological University [6]National University of Singapore [7]Vector Institute. Correspondence to: Yingru Li <szrlee@gmail.com>.

*Proceedings of the 43rd International Conference on Machine Learning*, Seoul, South Korea. PMLR 306, 2026. Copyright 2026 by the author(s).

In this work, we project the global variance-minimization objective into a causal form to derive the **Optimal Token Baseline (OTB)**. The key to OTB lies in its use of accumulated gradient magnitude as the weights for each token. Since the gradient variance of a specific token is not an isolated value but rather an accumulation of stochasticity from the start of the sequence, OTB explicitly accounts for this accumulated noise to achieve effective variance reduction. Our contributions are summarized as follows:

- **Establish the Optimal Token Baseline:** We propose OTB to ensure stable training and extreme sampling efficiency, achieving high performance with smaller group sizes ($N = 4$) and reducing the total token budget.
- **Propose the Logit-Gradient Proxy:** We introduce a computationally "free" proxy that estimates the gradient norm using only forward-pass probabilities without requiring additional backward passes.
- **Provide Theoretical Guarantees:** We conduct rigorous analysis on the unbiasedness and variance reduction effects of our OTB, which provide solid theoretical support beyond the empirical demonstration.

## 2 Related Works

Training stability is a critical cornerstone of reliable RL algorithms for LLMs. Two main factors that cause training instability are the bias and variance inherent in gradient estimates used for policy updates. Importance sampling (Yao et al., 2025; Liu et al., 2025) and trust region mask (Li et al., 2025) are widely adopted techniques to mitigate policies mismatch bias—a common source of instability arising from discrepancies between behavior and training policies—thus improving the stability of RL training.

Variance reduction represents another key avenue for enhancing training stability. Applying the learning rate decay scheduler (Yaxiang et al., 2025) can mitigate the signal-to-noise ratio, thereby facilitating variance reduction. Another prevalent strategy is to apply a baseline for advantage estimation, which helps dampen the stochasticity of gradient signals. Value functions (Schulman et al., 2015; 2017) are widely used for variance reduction in traditional RL; however, learning a value model is notoriously challenging in LLM-RL, due to the high dimensionality of token vocabularies. As an alternative, group-based baselines (Shao et al., 2024; Ahmadian et al., 2024; Zheng et al., 2025) have been proposed, which compute a baseline from the average reward of a group of sequences. A critical limitation of these methods lies in their neglect of sequence heterogeneity: they treat all sequences within a group as equally informative, failing to account for the varying gradient variance contributions of individual sequences.

To address sequence heterogeneity, prior work (Dayan, 1991; Greensmith et al., 2004; Weaver & Tao, 2013) has

formalized the Optimal Global Baseline (OGB). It weights reward expectations by the total gradient magnitude of individual sequences, theoretically achieving global variance reduction. Nevertheless, OGB relies on expensive gradient-based calculations, which hinders its practical deployment. Optimal Reward Baseline (OPO) (Hao et al., 2025) attempts to address this inefficiency by using sequence length as a proxy for gradient approximation. Another critical flaw of OGB is the neglect of token heterogeneity: tokens generated at the same step often exhibit different gradient magnitudes across distinct sequences. Additionally, OGB is inherently acausal as the total gradient magnitude of a sequence can only be computed after full sequence generation.

To address these limitations, we derive the causal Optimal Token Baseline, a dynamic mechanism for variance reduction. Using our Logit-Gradient Proxy, we efficiently estimate accumulated gradient variance of each token, enabling inverse weighting of updates.

## 3 Gradient Variance for Training Instability

This instability of RL in long-horizon LLM alignment is not merely a tuning issue, but a structural consequence of how *gradient variance scales with trajectory length and reward sparsity*. This can be analyzed from the lens of the standard REINFORCE (Williams, 1992) gradient estimator: $\hat{g}(\tau) = R(\tau) \cdot S(\tau)$, where $\tau = (x, y)$ is the trajectory comprising the prompt $x$ and the generated response $y = (y_1, y_2, \ldots, y_T)$. The horizon length $T = |y|$ is a random variable. $R(\tau)$ is the total reward and $S(\tau) = \sum_{t=1}^{T} s_t$ is the total trajectory score, where $s_t = \nabla_\theta \log \pi_\theta(y_t \mid x, y_{<t})$. The noise in the gradient is mainly driven by two factors:

- **Long Horizons Accumulate Noise:** $S(\tau)$ is a sum of random variables. As the horizon $T$ grows, the variance of this sum increases like a random walk.
- **Sparse Rewards Amplify Noise:** $R(\tau)$ is observed at the very end, resulting in every step $t$ being scaled by the same, potentially high-variance, final outcome.

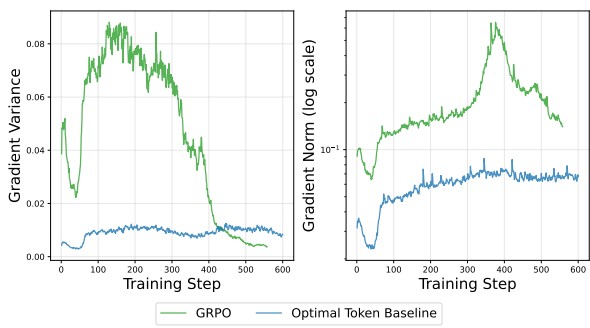

*Figure 2.* High gradient variance triggers a sudden surge in the gradient norm, leading to an eventual training collapse. The calculation of gradient variance is introduced in Appendix E.

A common approach to reduce gradient variance is to introduce a baseline $B(x)$, yielding $\tilde{g}(\tau) = (R(\tau) - B(x)) \cdot S(\tau)$. Group-mean baselines are widely adopted for LLM-RL, which apply the empirical estimator of the expected reward as the baseline: $B(x) = \mathbb{E}_\tau[R(\tau)]$, aiming to normalize rewards across sequences. These methods operate on the assumption that every sequence within a group is equally informative. However, they still suffer from large gradient variance and training collapse, as shown in Figure 2.

The key reason for the failure of group-mean baselines stems from their neglect of **sequence heterogeneity**. Specifically, sequences in the same group exhibit varying levels of **Total Energy**, which we define as $\|S(\tau)\|^2$, as shown in Figure 3.

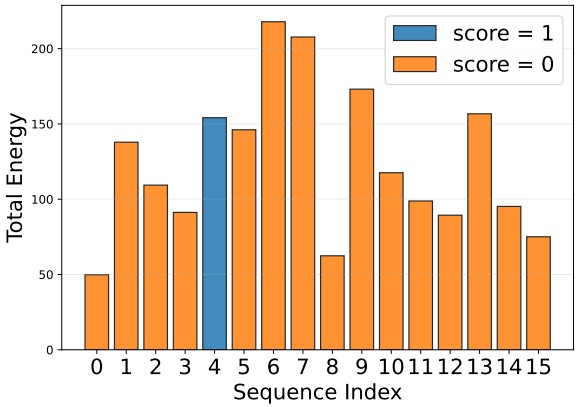

*Figure 3.* Different sequences in the same group exhibit distinct energy. The calculation of total energy is provided in Appendix A.2.

To address sequence heterogeneity, previous work (Dayan, 1991; Greensmith et al., 2004; Weaver & Tao, 2013; Li et al., 2023) has applied the principle of variance reduction and derived the Optimal Global Baseline (OGB). It is the reward expectation weighted by the total energy $\|S(\tau)\|^2$:

$$B^*_{global}(x) = \frac{\mathbb{E}_\tau \left[ R(\tau) \cdot \|S(\tau)\|^2 \right]}{\mathbb{E}_\tau \left[ \|S(\tau)\|^2 \right]}. \quad (1)$$

It points out that sequences with total score $S(\tau)$ contribute quadratically more to the variance. The baseline $B^*_{global}(x)$ must fit these high-energy sequences most accurately to prevent instability. We provide the detailed analysis on OGB for trajectory-level variance reduction in Appendix A.1.

However, OGB remains suboptimal as it overlooks **token heterogeneity**. Specifically, tokens at the same generation step $t$ contribute varying levels of energy across different sequences. Therefore, the total energy fails to accurately represent the impact of individual tokens. To quantify the energy cumulation effect up to time step $t$, we define the **Realized Energy** as follows:

**Definition 3.1** (Realized Energy).

$$W_t \equiv \sum_{j=1}^{t} \|s_j\|^2 = \sum_{j=1}^{t} \|\nabla_\theta \log \pi_\theta(y_j \mid x, y_{<t})\|^2. \quad (2)$$

This realized energy serves as a stable proxy for the sequence's accumulated instability, effectively measuring the total gradient magnitude expended up to the current step. As illustrated in Figure 4, the energy profile of a sequence is not static; sequences that appear low energy at step $t$ may surge in energy by step $t + k$. Since the rankings of realized energy across sequences frequently intersect over time, the total energy of each sequence cannot adequately characterize the impact of each token.

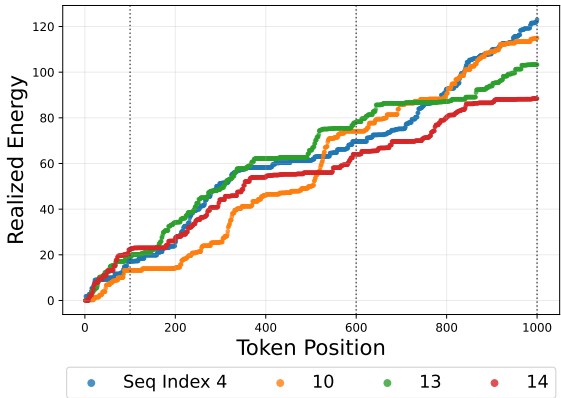

*Figure 4.* Individual tokens contribute varying energy. Within a single generation step $t$, sequences exhibit distinct energy profiles. For instance, at $t = 100$, the realized energy ranks from lowest to highest as Seq $10 < 4 < 13 < 14$. However, this ranking shifts significantly by $t = 600$: the order becomes Seq $14 < 4 < 10 < 13$, with further re-ranking occurring by $t = 1000$. The calculation of realized energy is provided in Section 4.1.

Another flaw of OGB is inherent **acausality**. Namely, we cannot know the total energy $\|S(\tau)\|^2$ at the intermediate token generation step $t$.

## 4 The Optimal Token Baseline

To address these limitations and achieve further gradient variance reduction, we first switch to the causal policy gradient (Williams, 1992) (see details in Appendix B). Specifically, we introduce the reward-to-go $G_t = \sum_{k=t}^{T} r_k$, and a time-dependent baseline sequence $\{B_t\}_{t=1}^{T}$. This yields the causal gradient estimator $\tilde{g}_c(\tau) = \sum_{t=1}^{T} s_t(G_t - B_t)$.

Our main goal is to minimize the gradient variance:

$$\begin{aligned} \mathbb{V}[\tilde{g}_c(\tau)] &\triangleq \mathbb{E}_\tau \left[ \|\tilde{g}_c(\tau) - \mathbb{E}_\tau[\tilde{g}_c(\tau)]\|^2 \right] \quad (3) \\ &= \mathbb{E}_\tau \left[ \|\tilde{g}_c(\tau)\|^2 \right] - \|\mathbb{E}_\tau[\tilde{g}_c(\tau)]\|^2 \\ &= \mathbb{E}_\tau \left[ \|\tilde{g}_c(\tau)\|^2 \right] - \|\nabla_\theta \mathcal{J}(\theta)\|^2, \end{aligned}$$

where $\nabla_\theta \mathcal{J}(\theta)$ is the true gradient and is independent of the baselines. Thus minimizing the gradient variance is mathematically equivalent to minimizing the expected squared norm of the gradient estimator $\mathbb{E}_\tau[\|\tilde{g}_c(\tau)\|^2]$.

To derive each optimal time-dependent baseline $B_t$, we apply the realized energy $W_t$ from Definition 3.1 and formulate the following per-time-step optimization objective:

$$\min_{B_t} \mathcal{J}(B_t) = \mathbb{E}_{y_{\leq t} \sim \pi_\theta(\cdot|x)} \left[ \sum_{j=1}^t \|s_j\|^2 (G_t - B_t)^2 \right] \quad (4)$$
$$= \mathbb{E}_{y_{\leq t} \sim \pi_\theta(\cdot|x)} \left[ W_t (G_t - B_t)^2 \right],$$

where the full derivation and its connection to the computationally intractable objective are deferred to Appendix C. By solving this objective, we obtain the Optimal Token Baseline under specific structural conditions:

**Theorem 4.1** (Optimal Token Baseline). *The Optimal Token Baseline (OTB) at step $t$ is the weighted centroid of the reward-to-go, weighted by the realized energy:*

$$B_t^*(x) = \frac{\mathbb{E}_{y_{\leq t} \sim \pi_\theta(\cdot|x)}[G_t \cdot W_t]}{\mathbb{E}_{y_{\leq t} \sim \pi_\theta(\cdot|x)}[W_t]}. \quad (5)$$

The key to OTB lies in using realized energy as the weights for each token, as the gradient variance of a specific token is not an isolated value but rather an accumulation of stochasticity from the start of the sequence. OTB accounts for this accumulated noise for variance reduction.

### 4.1 Logit-Gradient Proxy

Computing the squared norm of the true parameter gradient, $\|s_t\|^2$, requires a separate backward pass for each generated token, which is computationally prohibitive for LLMs. To implement OTB efficiently, we introduce the **Logit Gradient Norm** as a computable proxy to approximate the full parameter gradient norm. We assume the norm of the update to the whole parameters $\theta$ is proportional to the norm of the update to the logits $z_t$ and provide the justification in Section 5.3. Formally, the gradient of the log-likelihood with respect to the logits $z_t \in \mathbb{R}^{|\mathcal{V}|}$ is given by:

$$\delta_t \equiv \nabla_{z_t} \log \pi_\theta(y_t|x, y_{<t}) = \mathbf{e}_{y_t} - \boldsymbol{\pi}_t, \quad (6)$$

where $\boldsymbol{\pi}_t \in \Delta^{|\mathcal{V}|}$ represent the probability distribution vector over the vocabulary $\mathcal{V}$, and $\mathbf{e}_{y_t}$ is the one-hot encoding of the token $y_t$ sampled at step $t$. Based on this, we propose the Logit-Gradient Proxy as the squared norm of $\delta_t$:

**Proposition 4.2** (Logit-Gradient Proxy). *We use the Logit Gradient Norm as a proxy and derive a closed-form:*

$$\hat{w}_t = \|\delta_t\|^2 \equiv (\mathbf{e}_{y_t} - \boldsymbol{\pi}_t)^\top (\mathbf{e}_{y_t} - \boldsymbol{\pi}_t) \quad (7)$$
$$= \mathbf{e}_{y_t}^\top \mathbf{e}_{y_t} - 2\mathbf{e}_{y_t}^\top \boldsymbol{\pi}_t + \boldsymbol{\pi}_t^\top \boldsymbol{\pi}_t$$
$$= 1 - 2\pi_\theta(y_t \mid x, y_{<t}) + \|\boldsymbol{\pi}_t\|_2^2.$$

*The $\|\boldsymbol{\pi}_t\|_2^2 = \sum_{v \in \mathcal{V}} \pi_\theta^2(v \mid x, y_{<t})$ denotes the sum of squared token probabilities.*

This proxy requires zero additional backward-pass and only depends on the forward-pass output probabilities. It can effectively capture model confidence, as shown in Figure 5. The high-confidence tokens where $\pi_\theta(y_t \mid x, y_{<t}) \to 1$ have $\hat{w}_t \approx 0$ corresponding to low weights, and vice versa.

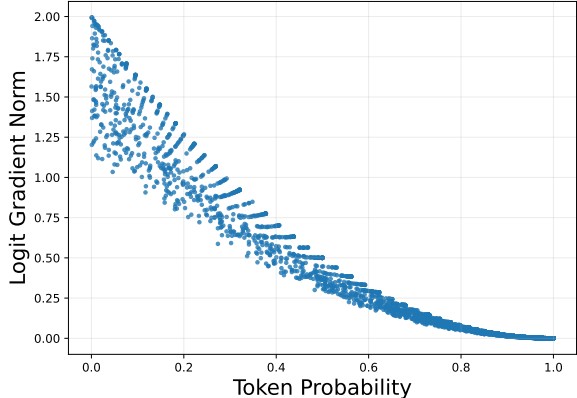

*Figure 5.* The relationship between Token Probability (Model Confidence) and Logit Gradient Norm (Uncertainty Measure).

Leveraging the Logit-Gradient Proxy, we can efficiently compute the practical OTB with a group of $N$ responses using the approximated realized energy $\hat{W}_t = \sum_{j=1}^t \hat{w}_j$:

$$\hat{B}_t = \frac{\sum_{i=1}^N G_t^{(i)} \cdot \hat{W}_t^{(i)}}{\sum_{i=1}^N \hat{W}_t^{(i)}}. \quad (8)$$

We demonstrate the workflow for calculating the OTB in Figure 6. Unlike standard group-mean baselines, which average over the padding *EOS* token ineffectively, OTB handles this naturally by computing baselines only over valid tokens.

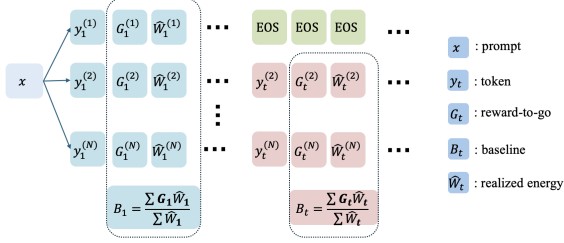

*Figure 6.* An overview of the Optimal Token Baseline.

Thus, OTB can dynamically assign distinct advantages to each token in the sequence through a value-model-free approach, as shown in Figure 7. This tracks the reward structure for different tokens, which cannot be achieved by Optimal Global Baseline or other group-mean baselines.

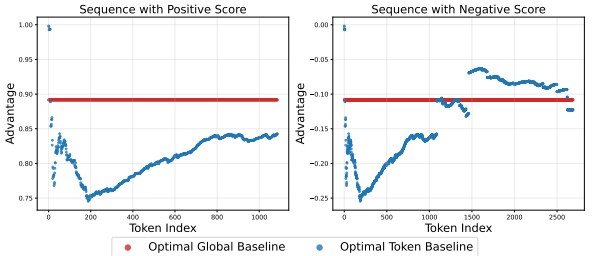

*Figure 7.* Advantage distribution comparison on positive and negative sequences from the same group. The Optimal Token Baseline dynamically adjusts the advantage for each token, in contrast to the OGB, which assigns an equivalent advantage across all tokens.

## 5 Theoretical Analysis

We now analyze the unbiasedness of the OTB, discuss its effectiveness in variance reduction, and present the justification of our Logit-Gradient Proxy.

### 5.1 Unbiasedness of the OTB

A fundamental requirement for any baseline is that it must not introduce bias into the policy gradient estimate. To ensure unbiasedness, the baseline at step $t$ must be independent of the sampled action $y_t$, meaning it can only depend on the trajectory history $y_{<t}$ and the prompt $x$. Our proposed OTB satisfies this condition because $B_t^*$ relies strictly on the prompt $x$, the current timestep $t$, and expectations taken over the policy distribution, as established in Theorem 4.1. Consequently, it remains completely independent of the specific realization of the token $y_t$ being evaluated in the current trajectory.

Mathematically, because $B_t^*$ is a scalar constant with respect to the token distribution at step $t$, then we have:

$$\mathbb{E}_{y_t}[\tilde{g}_c(y_t) \mid x, y_{<t}] \qquad (9)$$
$$=\mathbb{E}_{y_t}[s_t(G_t - B_t^*) \mid x, y_{<t}]$$
$$=\mathbb{E}_{y_t}[s_t \cdot G_t \mid x, y_{<t}] - \mathbb{E}_{y_t}[s_t \cdot B_t^* \mid x, y_{<t}]$$
$$=\nabla_\theta \mathcal{J}(\theta) - B_t^* \cdot \underbrace{\mathbb{E}_{y_t}[s_t \mid x, y_{<t}]}_{0} = \nabla_\theta \mathcal{J}(\theta).$$

Similarly, the approximation $\hat{B}_t$ is computed by using group statistics. While group-dependent baselines (like the group mean in GRPO) introduce a negligible finite-sample bias of order $O(1/N)$, they remain asymptotically unbiased and are standard practice in RL. OTB adheres to this same standard, ensuring the optimization landscape remains valid.

Since $B_{t+1}^*$ is independent of the current action $y_t$, it technically remains a valid, unbiased candidate for the current step; however, it fails to minimize variance due to structural discrepancies. The primary limitation is a target mismatch, where $B_{t+1}^*$ approximates the future return $G_{t+1}$ rather than the target return $G_t$, leaving the immediate reward $r_t$ unnormalized and inflating variance in dense reward settings. Crucially, this is compounded by a causal weighting mismatch regarding the realized energy weights: while $B_t^*$ is specifically derived to minimize variance with respect to the cumulative weights $W_t$ observed up to step $t$, $B_{t+1}^*$ is optimized for the subsequent weights $W_{t+1}$, rendering it suboptimal for the current gradient estimate.

Consider a trajectory that is confident at step $t$ (low $W_t$) but becomes highly chaotic at step $t+1$ (high $W_{t+1}$). If we used $B_{t+1}^*$ at step $t$, we would be allowing the future instability at $t+1$ to distort the baseline at step $t$. This violates the causal structure of variance reduction. We must normalize the gradient at step $t$ based only on the accumulated uncertainty that has effectively "caused" the current variance. Using future weights would solve the wrong optimization problem, leading to a suboptimal baseline.

### 5.2 Variance Reduction of the OTB

In Section 4, we reformulate the gradient variance $\mathbb{V}[\tilde{g}_c(\tau)]$ minimization problem as minimizing the expected squared norm of the gradient estimator $\mathbb{E}_\tau[\|\tilde{g}_c(\tau)\|^2]$, and formalize the objective $\mathcal{J}(B_t)$ in Eq. (4) to identify the optimal $B_t$. To derive this optimal value, we take the derivative of $\mathcal{J}(B)$ with respect to $B_t$ and set the resulting expression to zero:

$$\frac{d\mathcal{J}(B_t)}{dB_t} = -2\mathbb{E}_{y_{\le t} \sim \pi_\theta(\cdot \mid x)}[W_t(G_t - B_t)] = 0. \quad (10)$$

Solving Eq. (10) yields the optimal baseline $B_t^*$, which matches exactly the result presented in Theorem 4.1. This confirms that $B_t^*$ is the exact solution to the gradient variance minimization problem. It effectively "dampens" the noise from high-uncertainty trajectories by centering the gradient estimate more aggressively on them.

We then analyze the superiority of the Optimal Token Baseline over OGB by comparing the objective value $\mathcal{J}$ of the static $B_{global}^*$ (omitting $x$ in $B_{global}^*(x)$ for brevity) against the dynamic OTB process $\mathbf{B}^* = \{B_t^*\}_{t=1}^T$:

$$\mathcal{J}(B_{global}^*) = \sum_t \mathbb{E}[W_t(G_t - B_{global}^*)^2] \qquad (11)$$
$$= \sum_t \mathbb{E}\left[W_t\left((G_t - B_t^*) + (B_t^* - B_{global}^*)\right)^2\right]$$
$$= \mathcal{J}(\mathbf{B}^*) + \text{Term A} + \text{Term B}.$$

The Term $A = 2 \sum_t \mathbb{E}[W_t(G_t - B_t^*)(B_t^* - B_{global}^*)]$ is the cross-term, which vanishes due because $\mathbb{E}[W_t(G_t - B_t^*)] = 0$ by the definition of $B_t^*$ from Eq. (10).

The Term $B = \sum_t \mathbb{E}[W_t(B_t^* - B_{global}^*)^2]$ corresponds to the variance gap. This gap is strictly positive as the $B_t^*$ evolves dynamically over time as show in Figure 7. In long-horizon tasks, $B_t^*$ changes drastically as the agent generates key reasoning steps, which a static baseline $B_{\mathrm{global}}^*$ is inherently unable to track. Therefore, the dynamic OTB consistently achieves lower gradient variance than the static OGB.

### 5.3 Justification for Logit-Gradient Proxy

We justify that the square norm of logit gradient $\|\delta_t\|^2$ is a valid substitute for the square norm of the true parameter gradient $\|\nabla_\theta\|^2$ by showing they are mathematically proportional for modern Transformer architectures.

Consider the final linear layer (Unembedding Head) $z_t = W h_t$, where $W$ is the weight matrix and $h_t$ is the hidden state. The gradient of the objective $J$ with respect to $W$ is given by the outer product of the logit error $\delta_t$ and the input $h_t$: $\nabla_W J = \delta_t h_t^\top$. The Frobenius norm of this rank-1 matrix decomposes into the product of the vector norms:

$$\|\nabla_W J\|_F = \sqrt{\mathrm{Tr}(\delta_t h_t^\top h_t \delta_t^\top)} = \|\delta_t\|_2 \cdot \|h_t\|_2. \quad (12)$$

In modern LLMs (Grattafiori et al., 2024; Bai et al., 2023), the hidden state $h_t$ is the output of an RMSNorm layer. RMSNorm normalizes the input vector $x$ such that its root mean square is constant:

$$h_t = \mathrm{RMSNorm}(x) \implies \|h_t\|_2 \approx \sqrt{d_{model}}. \quad (13)$$

Since $\|h_t\|_2$ is approximately constant across all time steps, the gradient norm of the last layer is strictly proportional to the logit gradient norm: $\|\nabla_W J\|_F \propto \|\delta_t\|_2$. To extend this justification to the full parameter gradient $s_t$, we adopt the standard assumption that the backbone network propagates gradients isotropically. Under this assumption, the proportionality observed in the final layer is preserved throughout the backward pass, implying $\|s_t\|^2 \propto \|\delta_t\|^2$.

Since the Optimal Token Baseline is a weighted centroid (a ratio), any constant scalar factor cancels out. Thus, using the logit gradient proxy is mathematically equivalent to using the true gradient norm of the final layer.

## 6 Empirical Studies

We conduct comprehensive experiments to evaluate Optimal Token Baseline across two distinct paradigms for mathematical tasks with rule-based reward signals: Single-Turn Reasoning and Multi-Turn Tool-Integrated Reasoning (TIR). For the TIR setting, we adhere to the setup from Simple-TIR (Xue et al., 2025), requiring LLM generates Python code throughout a maximum of 5 turns. For both paradigms, we adopt full on-policy training on the unaligned Qwen series base models, using the deduplicated DAPO-MATH-17k as the training dataset. We remain hyperparameters remain consistent across both paradigms, including the batch size of 128, the maximum response length of 8192 and the group size of 16. For evaluation, we set the Top-p to 0.95 and report avg@32 scores to ensure statistically robust results. Further detailed settings are provided in Appendix F.1.

### 6.1 Comparative Results on Performance Metrics

We compare Optimal Token Baseline against several competitive methods, including GRPO (Shao et al., 2024), RLOO (Ahmadian et al., 2024), OPO (Hao et al., 2025), and OGB. The OPO is also derived from optimal baseline theory, yet it uses sequence length as a proxy to estimate gradient norm for trajectory-level variance reduction. We will discuss the limitations of this approximation strategy in Section 6.4. The implementation details of OGB are provided in Appendix A.2. For the TIR setting, we incorporate SimpleTIR (Xue et al., 2025)—a strong method that filters invalid trajectories for training stability.

*Table 1.* Comparative performance results on mathematical reasoning benchmarks. All scores represent the peak performance achieved by each method during the full training process.

| Method | AIME25 | AIME24 | AMC23 | MATH500 |
|---|---|---|---|---|
| *Single-Turn Reasoning on Qwen3-8b-Base* | | | | |
| GRPO | 25.31 | 31.35 | 80.86 | 90.50 |
| RLOO | 24.38 | 28.96 | 74.53 | 90.56 |
| OPO | 27.29 | 35.31 | 84.53 | 93.31 |
| OGB | 30.10 | 33.13 | 80.55 | 90.75 |
| **OTB** | **30.31** | **37.29** | **85.08** | **93.43** |
| *TIR on Qwen2.5-7B* | | | | |
| GRPO | 18.54 | 27.60 | 59.45 | 76.88 |
| RLOO | 20.10 | 25.00 | 62.42 | 76.06 |
| OPO | 20.83 | 29.90 | 64.84 | 76.94 |
| OGB | 21.15 | 30.63 | 62.50 | 75.69 |
| SimpleTIR | 26.67 | 37.91 | 71.25 | 82.25 |
| **OTB** | **28.13** | **41.46** | **79.45** | **84.69** |

As shown in Table 1, our OTB achieves the best performance consistently across all settings. These findings indicate that training stability is the key factor to unlocking the full potential of LLM-RL. We further provide training curves for all methods in Appendix F.2, which demonstrate that OTB achieves notable stable training.

## 6.2 Eliminating Training Collapse

We have demonstrated that Optimal Token Baseline can effectively reduce gradient variance, leading to stable gradient norms and training dynamics, as shown in Figures 1 and 2. A secondary benefit of this stability is the mitigation of "Training-Rollout Mismatch"—a prevalent challenge in LLM-RL characterized by the drift between the policy being trained and the policy collecting rollout data. This mismatch introduces gradient bias, thereby degrading policy optimization and causing training collapse. As illustrated in Figure 8, OTB maintains minimal KL divergence between the training and rollout policies and preserves a steady response length, thereby extending the stable training window. Additionally, we provide supplementary metrics (e.g., policy entropy) in the Appendix F.3 to further validate the stability of OTB.

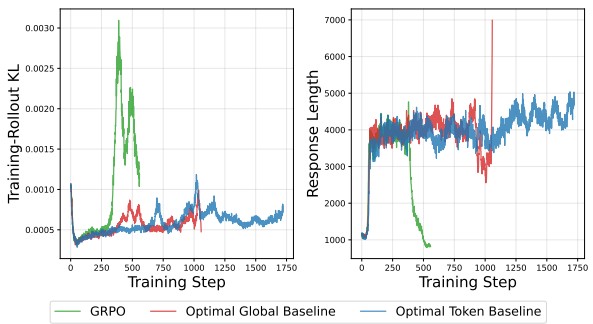

*Figure 8.* Training-Rollout KL Divergence and Response Length under Single-Turn Reasoning.

Other widely adopted strategies for enhancing training stability are *Truncated Importance Sampling* (TIS) (Yao et al., 2025) and *Masked Importance Sampling* (MIS) (Liu et al., 2025). These methods are designed to mitigate the mismatch bias induced by the discrepancies between the training policy and the behavior policy. To evaluate the efficacy of our Optimal Token Baseline, we compare it against these established techniques. We implement token-level TIS and MIS within the GRPO. As illustrated in Figure 9, our empirical findings suggest that while TIS and MIS alleviate instability in Single-Turn Reasoning tasks, they fundamentally struggle in TIR scenarios. In contrast, our OTB achieves consistent stability across all scenarios.

This finding suggests that instability in LLM-RL is not driven solely by policy divergence, but also by fundamental optimization challenges related to variance. By explicitly reducing gradient variance, OTB stabilizes parameter updates and drives the model toward a robust equilibrium. This effectively minimizes policies mismatch without the information loss associated with masking, a conclusion further corroborated by the empirical findings in Figure 8.

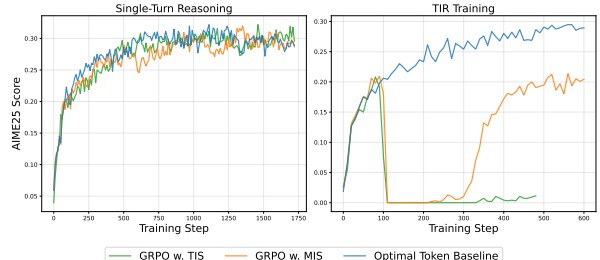

*Figure 9.* The Optimal Token Baseline demonstrates superior training stability compared to TIS and MIS.

## 6.3 Breaking the Sample Efficiency Barrier

Sample efficiency is the primary bottleneck of on-policy RL. Standard methods rely on large group size to average out noise and will suffer from early training collapse with a smaller group size. Because OTB assigns credit precisely by downweighting "noisy" tokens via the realized energy, it extracts significantly more signal per sample. As shown in Figure 10, OTB maintains training stability even under high-gradient-variance conditions (small group size $N$), proving that its variance reduction is sufficient to prevent training callospe. Specifically, OTB achieves comparable performance with a small group size of $N = 4$ as with a substantially larger group size of $N = 32$. This breakthrough enables stable training while **reducing total token consumption by 66.03% for Single-Turn Reasoning and 68.32% for TIR training**.

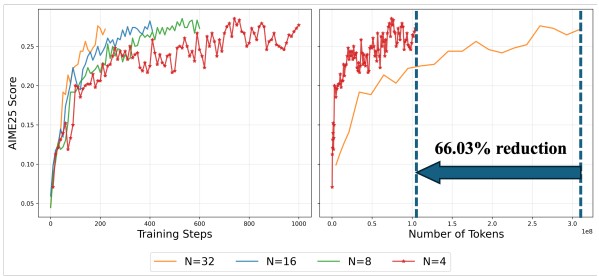

(a) Results under Single-Turn Reasoning.

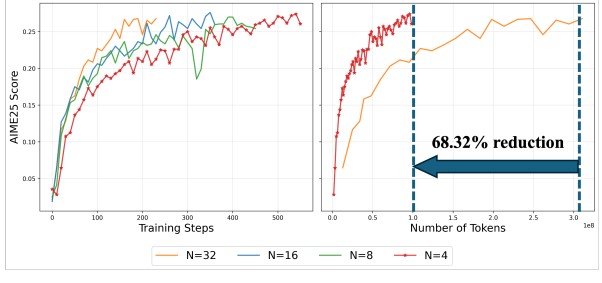

(b) Results under TIR.

*Figure 10.* Ablation study of group size $N$. Evaluated under the training step (Left) and total token budget (Right), OTB maintains high performance even with a minimal group size of $N = 4$.

## 6.4 The Logit-Gradient Proxy Matters

Another approach based on optimal baseline theory, OPO (Hao et al., 2025), operates on the assumption that longer sequences inherently entail higher variance (represented as larger energy) and should therefore be downweighted.

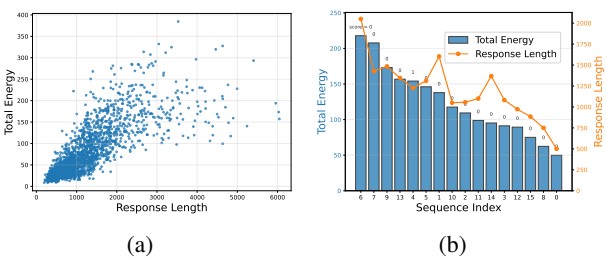

(a)                              (b)

*Figure 11.* Total energy and response length per sequence under (a) global batch distribution and (b) local intra-group distribution.

However, our empirical analysis in Figure 11 refutes this simplification. From a batch-wide perspective, we observe that sequences of identical response length exhibit widely varying levels of total energy. More critically, at the intra-group level, longer sequences also possess lower energy. This indicates that length-based approximations fail to capture the actual gradient variance inherent in each sequence.

Additionally, this **Length Proxy** fails to address token heterogeneity. Since the sequence length is identical for all sequences at any given generation step $t$, this metric assigns uniform importance to every token. This prevents the assignment of distinct per-token advantages, except in cases where sequences terminate early as shown in Figure 12. Therefore, the method is inherently unable to capture fine-grained, token-level energy differences.

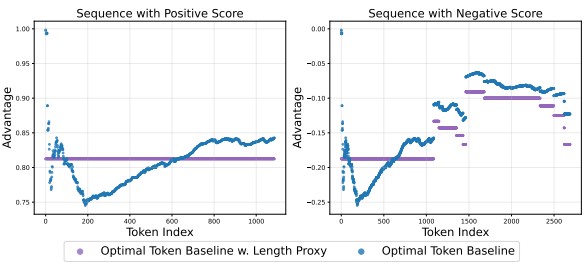

*Figure 12.* Advantage distribution comparison on positive and negative sequences from the same group. The Length Proxy cannot dynamically assign distinct per-token advantages unless certain sequences terminate generation with padded tokens.

We further evaluate the effectiveness of our Logit-Gradient Proxy by comparing it against the Length Proxy within both the Optimal Global Baseline and Optimal Token Baseline. As shown in Figure 13, relying on the Length Proxy causes both OGB and OTB to suffer from early training collapse. While integrating OGB with our Logit-Gradient Proxy extends the stable training duration, it eventually succumbs to collapse due to its inability to account for token heterogeneity. In contrast, OTB equipped with the Logit-Gradient Proxy effectively tracks the realized energy of each token generation step, ensuring sustained training stability.

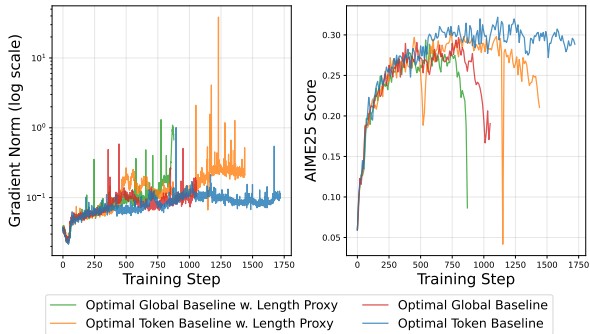

*Figure 13.* These results further confirm that Logit-Gradient Proxy is critical for training stability.

## 6.5 Robustness under Longer Contexts

We stress-test OTB in highly challenging scenarios by scaling the response length (from $8k$ to $16k$ tokens) and increasing the interaction turns (from 5 to 10 turns) within the TIR setting. As shown in Figure 14, OTB maintains consistent training stability across these demanding conditions. In the TIR setting, response lengths often exhibit substantial variance within a single group, rendering standard group mean baselines ineffective. OTB naturally accommodates this variability by dynamically computing baselines exclusively over valid tokens as shown in Figure 6.

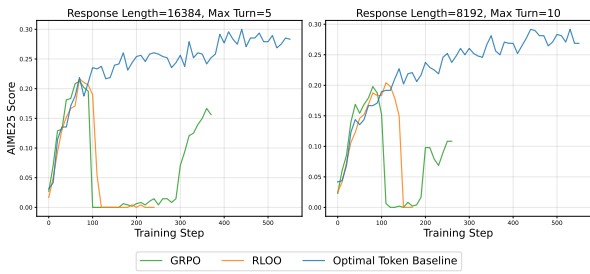

*Figure 14.* OTB consistently demonstrates stability under longer response length and larger maximum turn.

We additionally evaluate OTB on a large-scale model (14B) in Appendix F.4. Furthermore, we conduct a comparative analysis of an alternative variant of OTB that employs approximated isolated energy $\hat{w}_t$ to weight the reward-to-go in Appendix F.5. Experimental results demonstrate that our OTB consistently yields superior training stability and attains higher scores across all settings.

# 7 Conclusion and Discussion

We propose Optimal Token Baseline, which shifts from existing heuristic tricks to rigorous derivations in variance reduction for LLM-RL. We identify that not all tokens are created equal, while their contribution to the update must be weighted by their specific uncertainty, as quantified by the realized energy. This enables precise reward structure tracking by downweighting noisy tokens through a value-model-free approach. OTB achieves extreme sample efficiency: high performance is attained even with group sizes as small as $N = 4$. Crucially, we bridge the gap between theory and practice with the Logit-Gradient Proxy. This allows practitioners to estimate the true parameter gradient norm using only the logits from the forward pass, offering a *free lunch* for stability estimation.

In future work, we will explore the stability of OTB on more tasks such as search engines. As reasoning chains grow longer and agents become more autonomous, variance control becomes the defining challenge. We believe OTB provides the foundational stability required to scale LLM reasoning and agent to the next level of complexity.

## Acknowledgements

Baoxiang Wang and Jiawei Xu are partially supported by the National Natural Science Foundation of China (72394361) and the Shenzhen Science and Technology Program (JCYJ20250604141218024, JCYJ20250604141032005).

## Impact Statement

This work advances the RL training for LLM with a token-level advantage estimator, focusing on stable training and sample efficiency. There are many potential societal consequences of our work, none which we feel must be specifically highlighted here.

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

# A    Detailed Analysis on The Optimal Gloabl Baseline

In this section, we present a detailed derivation of the Optimal Global Baseline, demonstrate its trajectory-level variance reduction property, and then introduce our partial implementation.

## A.1    Trajectory-level Variance Reduction

In policy optimization, we often use a baseline $B(x)$—a function of the prompt $x$—to reduce the variance of the gradient estimate. The gradient estimator is defined as:

$$\tilde{g}(\tau) = (R(\tau) - B(x)) \cdot S(\tau). \tag{14}$$

This estimator remains unbiased for any choice of $B(x)$ because the expectation of the baseline term is zero:

$$\mathbb{E}_\tau[B(x) \cdot S(\tau)] = B(x) \cdot \mathbb{E}_\tau[S(\tau)] = 0. \tag{15}$$

Thus, $\mathbb{E}_\tau[\tilde{g}(\tau)] = \mathbb{E}_\tau[R(\tau) \cdot S(\tau)] = \nabla_\theta \mathcal{J}(\theta)$, confirming that the estimator remains centered on the true gradient for any choice of $B(x)$.

The goal of Optimal Global Baseline is to choose a baseline $B(x)$ to minimize the variance of this gradient estimator. The variance of the random vector $\tilde{g}(\tau)$ is the expected squared distance from its mean:

$$\mathrm{Var}(\tilde{g}(\tau)) = \mathbb{E}_\tau[\|\tilde{g}(\tau) - \nabla_\theta \mathcal{J}(\theta)\|^2]. \tag{16}$$

By expanding this expression, we get $\mathrm{Var}(\tilde{g}(\tau)) = \mathbb{E}_\tau[\|\tilde{g}(\tau)\|^2] - \|\nabla_\theta \mathcal{J}(\theta)\|^2$. Since the true gradient $\nabla_\theta \mathcal{J}(\theta)$ is independent of the baseline, minimizing variance is mathematically equivalent to minimizing the expected squared norm of the estimator. This leads to the following optimization objective to obtain the variance-minimizing:

$$\min_B \mathcal{J}(B) = \mathbb{E}_\tau[\|\tilde{g}(\tau)\|^2] = \mathbb{E}_\tau[\|(R(\tau) - B) \cdot S(\tau)\|^2]. \tag{17}$$

Expanding $\mathcal{J}(B)$ as a quadratic function of $B$ (using $B$ for $B(x)$ for brevity):

$$\begin{aligned}
\mathcal{J}(B) &= \mathbb{E}_\tau[(R(\tau) - B)^2 \cdot \|S(\tau)\|^2] \\
&= \mathbb{E}_\tau[R(\tau)^2 \cdot \|S(\tau)\|^2] - 2B\mathbb{E}_\tau[R(\tau)\|S(\tau)\|^2] + B^2\mathbb{E}_\tau[\|S(\tau)\|^2].
\end{aligned} \tag{18}$$

To find the minimum, we differentiate with respect to $B$ and set the result to zero:

$$\frac{d\mathcal{J}(B)}{dB} = -2\mathbb{E}_\tau[R(\tau)\|S(\tau)\|^2] + 2B\mathbb{E}_\tau[\|S(\tau)\|^2] = 0. \tag{19}$$

Solving for $B$ yields the Optimal Global Baseline:

$$B^*(x) = \frac{\mathbb{E}_\tau\left[R(\tau) \cdot \|S(\tau)\|^2\right]}{\mathbb{E}_\tau\left[\|S(\tau)\|^2\right]}. \tag{20}$$

The positivity of $\mathbb{E}_\tau[\|S(\tau)\|^2]$ confirms $\mathcal{J}(B^*(x))$ is a minimum. This result proves that to achieve minimum variance, the baseline must be a weighted average of rewards, where the weights are determined by the total energy $\|S(\tau)\|^2$.

### A.2 Practical Implementation

We use the Realized Energy (Definition 3.1) to define the total energy of a trajectory as $W(\tau)$. By applying our Logit-Gradient Proxy, we can efficiently approximate the total energy as $\hat{W}(\tau) = \sum_{j=1}^{T} \hat{w}_j$. The Optimal Global Baseline is then computed efficiently as:

$$\hat{B}_{\text{global}}(x) = \frac{\sum_{i=1}^{N} R^{(i)}(\tau) \cdot \hat{W}^{(i)}(\tau)}{\sum_{i=1}^{N} \hat{W}^{(i)}(\tau)}. \tag{21}$$

## B Derivation of the Causal Policy Gradient

We provide a detailed derivation of the causal policy gradient estimator (Williams, 1992). We demonstrate how the transition from a total trajectory reward $R(\tau)$ to a reward-to-go $G_t$ formulation reduces variance without introducing bias.

The objective in LLM alignment is to maximize the expected return $\mathcal{J}(\theta) = \mathbb{E}_{\tau \sim \pi_\theta}[R(\tau)]$. The standard REINFORCE gradient is given by:

$$\nabla_\theta \mathcal{J}(\theta) = \mathbb{E}_{\tau \sim \pi_\theta} \left[ R(\tau) \sum_{t=1}^{T} \nabla_\theta \log \pi_\theta(y_t \mid x, y_{<t}) \right]. \tag{22}$$

While unbiased, this estimator suffers from high variance because the total score function $\sum_{t=1}^{T} s_t = \sum_{t=1}^{T} \nabla_\theta \log \pi_\theta(y_t \mid x, y_{<t})$ acts as a single scalar multiplier for the total reward $R(\tau)$, ignoring the temporal structure of the generation.

In the context of autoregressive generation, a token $y_t$ at time $t$ can only affect subsequent rewards $\{r_t, r_{t+1}, \ldots, r_T\}$. It cannot physically influence rewards received at $k < t$. Mathematically, we can leverage the property that the expectation of the score function is zero: $\mathbb{E}_{y_t \sim \pi_\theta}[s_t] = 0$. Using the law of iterated expectations, we can show that:

$$\mathbb{E}_\tau[s_t r_k] = 0 \quad \text{for} \quad k < t. \tag{23}$$

By removing these **acausal** terms from the summation, we arrive at the causal policy gradient estimator:

$$\nabla_\theta \mathcal{J}(\theta) = \mathbb{E}_\tau \left[ \sum_{t=1}^{T} s_t \left( \sum_{k=t}^{T} r_k \right) \right] = \mathbb{E}_\tau \left[ \sum_{t=1}^{T} s_t G_t \right]. \tag{24}$$

The variance reduction in the causal policy gradient stems from the removal of acausal terms that contribute noise without providing useful signal. In the standard REINFORCE estimator, every score function $s_t$ is scaled by the total reward $R(\tau)$, meaning a token generated at the end of a sequence is erroneously scaled by rewards that occurred before it was even sampled. By substituting the total reward $R(\tau)$ with the reward-to-go $G_t = \sum_{k=t}^{T} r_k$, we effectively decouple each action from past rewards, which are mathematically equivalent to constants with respect to the current action. This is particularly critical for long-horizon LLM tasks; while the global estimator scales every step by a sum of $T$ random variables, the causal estimator scales the $t$-th step by only $T - t + 1$ variables.

Mathematically, if we consider rewards $r_t$ as random variables with variance $\sigma_r^2$, the variance of the reward term in REINFORCE remains at approximately $T\sigma_r^2$ for every token in the trajectory. In contrast, the causal estimator's reward variance decreases linearly as $t$ increases, effectively halving the total reward-related noise across the trajectory on average.

## C The Derivation of Optimal Token Baseline

In this section, we derive the Optimal Token Baseline by minimizing the variance of the causal policy gradient estimator. We define the time-dependent baseline sequence as $\{B_t\}_{t=1}^{T}$, yielding the causal gradient estimator $\tilde{g}_c(\tau) = \sum_{t=1}^{T} s_t(G_t - B_t)$.

### C.1 Variance Minimization Objective

Our goal is to minimize the gradient variance:

$$
\begin{aligned}
\mathbb{V}[\tilde{g}_c(\tau)] &\triangleq \mathbb{E}_\tau\left[\|\tilde{g}_c(\tau) - \mathbb{E}_\tau[\tilde{g}_c(\tau)]\|^2\right] \\
&= \mathbb{E}_\tau\left[\|\tilde{g}_c(\tau)\|^2\right] - \|\mathbb{E}_\tau[\tilde{g}_c(\tau)]\|^2 \\
&= \mathbb{E}_\tau\left[\|\tilde{g}_c(\tau)\|^2\right] - \|\nabla_\theta\mathcal{J}(\theta)\|^2.
\end{aligned}
\tag{25}
$$

Since the true policy gradient $\nabla_\theta\mathcal{J}(\theta)$ is independent of the baseline $B_t$, minimizing the variance is equivalent to minimizing the expected squared norm $\mathbb{E}_\tau[\|\tilde{g}_c(\tau)\|^2]$.

### C.2 First-Order Optimality Condition

We determine the optimal baseline by setting the derivative with respect to $B_t$ to zero:

$$
\begin{aligned}
\nabla_{B_t}\mathbb{E}_\tau[\|\tilde{g}_c\|^2] &= \nabla_{B_t}\mathbb{E}_\tau\left[\left\|\sum_{k=1}^T s_k(G_k - B_k)\right\|^2\right] \\
&= \mathbb{E}_\tau\left[2\left\langle\sum_{k=1}^T s_k(G_k - B_k), \frac{\partial}{\partial B_t}(s_t(G_t - B_t))\right\rangle\right] \\
&= -2\mathbb{E}_\tau\left[\langle\tilde{g}_c, s_t\rangle\right].
\end{aligned}
\tag{26}
$$

Setting this to zero requires $\mathbb{E}_\tau[\langle\tilde{g}_c, s_t\rangle] = 0$. Expanding the inner product:

$$
\mathbb{E}_\tau[\langle\tilde{g}_c, s_t\rangle] = \sum_{k=1}^T \mathbb{E}_\tau[(G_k - B_k)\langle s_k, s_t\rangle] = 0.
\tag{27}
$$

Separating the diagonal term ($k = t$) from the cross-temporal terms ($k \neq t$):

$$
\mathbb{E}_\tau\left[(G_t - B_t)\|s_t\|^2\right] + \sum_{k\neq t}\mathbb{E}_\tau[(G_k - B_k)\langle s_k, s_t\rangle] = 0.
\tag{28}
$$

Solving for $B_t$ yields the theoretical optimal baseline:

$$
B_t^* = \frac{\mathbb{E}_\tau[G_t\|s_t\|^2] + \sum_{k\neq t}\mathbb{E}_\tau[(G_k - B_k)\langle s_k, s_t\rangle]}{\mathbb{E}_\tau[\|s_t\|^2]}.
\tag{29}
$$

*Remark* C.1 (Role of $B_k$ Terms). Under the causal constraint $B_k = B_k(y_{\leq k})$, the baseline terms in the cross-correlations satisfy $\mathbb{E}_\tau[B_k\langle s_k, s_t\rangle] = 0$ for $k \neq t$, since the score function has zero conditional mean: $\mathbb{E}[s_j \mid y_{<j}] = 0$. Thus, Eq. (29) simplifies to:

$$
B_t^* = \frac{\mathbb{E}_\tau[G_t\|s_t\|^2] + \sum_{k\neq t}\mathbb{E}_\tau[G_k\langle s_k, s_t\rangle]}{\mathbb{E}_\tau[\|s_t\|^2]}.
\tag{30}
$$

However, we retain the $(G_k - B_k)$ form in our derivation because: (i) it directly mirrors the advantage terms in the gradient estimator $\tilde{g}_c = \sum_k s_k(G_k - B_k)$, and (ii) it provides clearer motivation for the Gradient Consistency approximation, which posits that advantage-weighted cross-correlations can be consolidated through cumulative energy.

This exact solution is computationally intractable: each score vector $s_t = \nabla_\theta\log\pi_\theta(y_t \mid y_{<t})$ has dimension equal to the parameter count $d$, and estimating the cross-correlations $\mathbb{E}[G_k\langle s_k, s_t\rangle]$ requires storing all $T$ score vectors. We therefore develop a tractable approximation.

### C.3 Gradient Consistency Approximation

To obtain a tractable baseline, we invoke the *Gradient Consistency* assumption (Gur-Ari et al., 2018), which is empirically well-supported in deep Transformer optimization. This assumption posits that gradient directions exhibit positive correlation across tokens within a sequence:

$$
\langle s_k, s_t\rangle \approx \alpha_{k,t}\|s_k\|\|s_t\|, \quad \text{with } \alpha_{k,t} > 0.
\tag{31}
$$

We further observe that advantage terms $(G_k - B_k)$ tend to be positively correlated within a trajectory: tokens along high-return trajectories collectively exhibit positive advantages, while tokens along low-return trajectories collectively exhibit negative advantages. This occurs because trajectory-level return variation typically dominates token-level variation in LLM-RL settings with sparse terminal rewards.

Under these conditions, the cross-temporal terms reinforce the diagonal term coherently. This motivates the approximation:

$$\sum_{k \neq t}(G_k - B_k)\langle s_k, s_t \rangle \approx (G_t - B_t)\sum_{k \neq t}\|s_k\|^2. \tag{32}$$

The intuition is that gradients at different positions push in similar directions, and the cumulative effect of cross-temporal correlations can be captured by the current advantage scaled by the total energy at other positions.

Substituting this approximation into Eq. (28):

$$\mathbb{E}_\tau\left[(G_t - B_t)\|s_t\|^2\right] + \mathbb{E}_\tau\left[(G_t - B_t)\sum_{k \neq t}\|s_k\|^2\right] \approx 0$$

$$\mathbb{E}_\tau\left[(G_t - B_t)\sum_{k=1}^T\|s_k\|^2\right] = 0$$

$$\mathbb{E}_\tau\left[(G_t - B_t) \cdot W_T\right] = 0, \tag{33}$$

where $W_T = \sum_{k=1}^T\|s_k\|^2$ denotes the total gradient energy over the trajectory.

Solving for $B_t$:

$$B_t^* = \frac{\mathbb{E}_\tau[G_t \cdot W_T]}{\mathbb{E}_\tau[W_T]}. \tag{34}$$

### C.4 Causal Approximation

The baseline in Eq. (34) requires the full trajectory energy $W_T$, which depends on future gradient norms $\{\|s_k\|^2\}_{k>t}$ unavailable at decision time $t$. We now develop a causal surrogate using only $W_t = \sum_{k=1}^t\|s_k\|^2$.

Decompose the total energy as $W_T = W_t + W_{>t}$, where $W_{>t} = \sum_{k>t}\|s_k\|^2$ is the future energy. We invoke the following assumption:

**Assumption C.2** (Future Energy Homogeneity). For trajectories sampled from the same prompt, the future energy $W_{>t}$ exhibits low variance:

$$W_{>t}^{(i)} \approx \bar{W}_{>t} \quad \text{for all samples } i \in \{1, \dots, N\}. \tag{35}$$

This assumption is reasonable because $W_{>t}$ aggregates gradient magnitudes over future tokens, which depend primarily on the remaining sequence length and model architecture. While the *directions* of future gradients vary with trajectory content, their *magnitudes* are largely determined by structural factors that are shared across trajectories from the same prompt.

Under Assumption C.2, we can characterize the relationship between the full-trajectory baseline and its causal counterpart. Define:

$$\hat{B}_t^{W_t} = \frac{\sum_{i=1}^N G_t^{(i)} W_t^{(i)}}{\sum_{i=1}^N W_t^{(i)}}, \qquad \bar{G}_t = \frac{1}{N}\sum_{i=1}^N G_t^{(i)}. \tag{36}$$

Here $\hat{B}_t^{W_t}$ is the causal energy-weighted baseline and $\bar{G}_t$ is the sample mean return (analogous to a value function baseline).

**Proposition C.3** (Convex Decomposition). *Under Assumption C.2, the full-trajectory baseline decomposes as:*

$$\hat{B}_t^{W_T} = \alpha_t \cdot \hat{B}_t^{W_t} + (1 - \alpha_t) \cdot \bar{G}_t, \tag{37}$$

*where the mixing coefficient is:*

$$\alpha_t = \frac{\bar{W}_t}{\bar{W}_t + \bar{W}_{>t}}, \qquad \bar{W}_t = \frac{1}{N}\sum_{i=1}^N W_t^{(i)}. \tag{38}$$

*Proof.* Under the assumption $W_{>t}^{(i)} \approx \bar{W}_{>t}$ for all $i$:

$$\hat{B}_t^{W_T} = \frac{\sum_{i=1}^N G_t^{(i)} W_T^{(i)}}{\sum_{i=1}^N W_T^{(i)}} = \frac{\sum_{i=1}^N G_t^{(i)} (W_t^{(i)} + \bar{W}_{>t})}{\sum_{i=1}^N (W_t^{(i)} + \bar{W}_{>t})}$$
$$= \frac{\sum_{i=1}^N G_t^{(i)} W_t^{(i)} + \bar{W}_{>t} \sum_{i=1}^N G_t^{(i)}}{\sum_{i=1}^N W_t^{(i)} + N\bar{W}_{>t}}. \tag{39}$$

Let $S_W = \sum_{i=1}^N W_t^{(i)}$ and $S_G = \sum_{i=1}^N G_t^{(i)}$. Then:

$$\hat{B}_t^{W_T} = \frac{S_W}{S_W + N\bar{W}_{>t}} \cdot \frac{\sum_{i=1}^N G_t^{(i)} W_t^{(i)}}{S_W} + \frac{N\bar{W}_{>t}}{S_W + N\bar{W}_{>t}} \cdot \frac{S_G}{N}$$
$$= \alpha_t \cdot \hat{B}_t^{W_t} + (1 - \alpha_t) \cdot \bar{G}_t, \tag{40}$$

where $\alpha_t = \frac{S_W}{S_W + N\bar{W}_{>t}} = \frac{\bar{W}_t}{\bar{W}_t + \bar{W}_{>t}}$.  $\square$

**Interpretation.** Proposition C.3 reveals that the optimal baseline interpolates between two components:

- $\hat{B}_t^{W_t}$: the energy-weighted baseline, which leverages the correlation between cumulative gradient energy and returns to achieve variance reduction;

- $\bar{G}_t$: the simple mean baseline, corresponding to the classical value function approach.

The mixing coefficient $\alpha_t = \frac{\bar{W}_t}{\bar{W}_t + \bar{W}_{>t}}$ increases monotonically with $t$, reflecting that more of the trajectory's discriminative information becomes available as generation progresses.

For our practical implementation, we adopt the causal energy-weighted baseline:

$$\hat{B}_t = \hat{B}_t^{W_t} = \frac{\sum_{i=1}^N G_t^{(i)} \cdot W_t^{(i)}}{\sum_{i=1}^N W_t^{(i)}}. \tag{41}$$

This corresponds to emphasizing the energy-weighted component of Eq. (37). The choice is motivated by two considerations:

1. **Causality**: The baseline uses only information available at decision time $t$.

2. **Discriminative power**: Unlike the mean baseline $\bar{G}_t$, the energy-weighted baseline assigns higher importance to high-gradient trajectories, directly addressing the variance structure identified by our analysis.

### C.5 Empirical Estimator

To compute the OTB efficiently, we apply the Gradient-Norm Proxy developed in Section 4.1. We approximate the cumulative energy as $\hat{W}_t = \sum_{j=1}^t \hat{w}_j$, where $\hat{w}_j$ is a computationally efficient proxy for $\|s_j\|^2$. This yields the final empirical estimator:

$$\hat{B}_t = \frac{\sum_{i=1}^N G_t^{(i)} \cdot \hat{W}_t^{(i)}}{\sum_{i=1}^N \hat{W}_t^{(i)}}. \tag{42}$$

This matches the expression in Theorem 4.1.

### C.6 Extension: Logit-Gradient Proxy for Cross Terms

In Section 4.1, we introduced the Logit-Gradient Proxy $\hat{w}_t = \|\delta_t\|^2$ to approximate the parameter gradient norm $\|s_t\|^2$, where $\delta_t = e_{y_t} - \pi_t$ is the logit-space gradient. A natural question is whether a similar approach can approximate the cross-correlation terms $\langle s_k, s_t \rangle$ that appear in the theoretical optimal baseline.

**Decomposition of Cross Terms.** Recall that the parameter-space score function can be written as:

$$s_t = J_t^\top \delta_t, \tag{43}$$

where $J_t = \nabla_\theta z_t \in \mathbb{R}^{|V| \times d}$ is the Jacobian of logits with respect to parameters. The cross-correlation between score functions at positions $k$ and $t$ is:

$$\langle s_k, s_t \rangle = \delta_k^\top J_k J_t^\top \delta_t = \delta_k^\top M_{k,t} \delta_t, \tag{44}$$

where $M_{k,t} = J_k J_t^\top \in \mathbb{R}^{|V| \times |V|}$.

For the diagonal case ($k = t$), our Logit-Gradient Proxy assumes $M_{t,t} = J_t J_t^\top \propto I$, yielding $\|s_t\|^2 \approx c \cdot \|\delta_t\|^2$. Extending this to the off-diagonal case, if we assume $M_{k,t} \approx \lambda_{k,t} I$ for some scalar $\lambda_{k,t}$, then:

$$\langle s_k, s_t \rangle \approx \lambda_{k,t} \cdot \langle \delta_k, \delta_t \rangle. \tag{45}$$

**Logit-Space Cross-Correlation.** The logit-space inner product $\langle \delta_k, \delta_t \rangle$ admits a closed-form expression:

$$\begin{aligned}
\langle \delta_k, \delta_t \rangle &= (e_{y_k} - \pi_k)^\top (e_{y_t} - \pi_t) \\
&= e_{y_k}^\top e_{y_t} - e_{y_k}^\top \pi_t - \pi_k^\top e_{y_t} + \pi_k^\top \pi_t \\
&= \mathbf{1}[y_k = y_t] - \pi_t(y_k) - \pi_k(y_t) + \langle \pi_k, \pi_t \rangle,
\end{aligned} \tag{46}$$

where $\mathbf{1}[y_k = y_t]$ is the indicator for token equality, $\pi_t(y_k)$ denotes the probability assigned to token $y_k$ at position $t$, and $\langle \pi_k, \pi_t \rangle = \sum_{v \in \mathcal{V}} \pi_k(v) \pi_t(v)$ is the dot product of probability distributions. Notably, this expression requires only forward-pass quantities.

**Challenges for Direct Application.** Unlike the diagonal term $\|\delta_t\|^2 \geq 0$, the cross term $\langle \delta_k, \delta_t \rangle$ can be negative. For instance, when $y_k \neq y_t$ and the model assigns significant probability to $y_k$ at position $t$ (i.e., $\pi_t(y_k)$ is large), the cross term becomes negative. This sign variability complicates direct substitution into the gradient consistency framework, which relies on positive reinforcement across time steps.

Furthermore, the scaling factor $\lambda_{k,t}$ in the approximation $M_{k,t} \approx \lambda_{k,t} I$ may vary significantly across position pairs, unlike the diagonal case where a single constant suffices. Estimating these factors would require additional computation that undermines the efficiency gains of the proxy approach.

**Justification for Cumulative Energy Approach.** These considerations provide additional motivation for our gradient consistency approximation (Eq. 32), which sidesteps direct computation of cross terms by consolidating their effect through cumulative energy weighting. The approximation:

$$\sum_{k \neq t} (G_k - B_k) \langle s_k, s_t \rangle \approx (G_t - B_t) \sum_{k \neq t} \|s_k\|^2 \tag{47}$$

effectively assumes that the net contribution of cross-temporal correlations is positive and proportional to the total energy at other positions. This is a stronger but more tractable assumption than modeling each $\langle s_k, s_t \rangle$ individually.

**Potential Extensions.** An alternative baseline variant could directly incorporate logit-space cross terms:

$$\tilde{W}_t = \sum_{k=1}^{t} \langle \delta_k, \delta_t \rangle^+, \tag{48}$$

where $(\cdot)^+ = \max(0, \cdot)$ clips negative values to maintain non-negative weights. We leave empirical investigation of such variants to future work, noting that our current approach using cumulative energy $\hat{W}_t = \sum_{k=1}^{t} \hat{w}_k$ already achieves strong performance while maintaining computational simplicity.

## C.7 Connections with Prior Work

Our derivation provides a unified view of existing baselines as special cases of the optimal baseline under progressively stronger assumptions.

**Value Function Baseline.**    Starting from the theoretical optimal baseline (Eq. 29):

$$B_t^* = \frac{\mathbb{E}_\tau[G_t\|s_t\|^2] + \sum_{k\neq t}\mathbb{E}_\tau[(G_k - B_k)\langle s_k, s_t\rangle]}{\mathbb{E}_\tau[\|s_t\|^2]}, \tag{49}$$

the classical value function baseline emerges under two simplifying assumptions:

1. **No cross-temporal correlation**: $\langle s_k, s_t\rangle = 0$ for $k \neq t$, which eliminates the second term;

2. **Deterministic score norm given prefix**: $\|s_t\|^2 = c_t(y_{<t})$ depends only on the prefix, not on the choice of $y_t$ or future tokens.

Under these assumptions, the optimal baseline conditioned on prefix $y_{<t}$ becomes:

$$B_t^*(y_{<t}) = \frac{\mathbb{E}[G_t\|s_t\|^2 \mid y_{<t}]}{\mathbb{E}[\|s_t\|^2 \mid y_{<t}]} = \frac{c_t(y_{<t}) \cdot \mathbb{E}[G_t \mid y_{<t}]}{c_t(y_{<t})} = \mathbb{E}[G_t \mid y_{<t}] = V(x, y_{<t}), \tag{50}$$

recovering the value function. Note that conditioning on the prefix $y_{<t}$ is essential for this reduction; without it, we would obtain $\mathbb{E}_\tau[G_t] = \mathbb{E}_{y_{<t}}[V(x, y_{<t})]$, the expected value function averaged over all prefixes, rather than the state-dependent value function itself. This reveals that traditional actor-critic methods implicitly assume both gradient independence across time and homogeneous gradient magnitudes at each state—assumptions that are violated in deep Transformer architectures.

**Isolated Energy Baseline.**    Relaxing only the second assumption (allowing $\|s_t\|^2$ to vary) while maintaining cross-temporal independence yields:

$$B_t^{\text{isolated}} = \frac{\mathbb{E}_\tau[G_t \cdot \|s_t\|^2]}{\mathbb{E}_\tau[\|s_t\|^2]}. \tag{51}$$

This baseline, studied in classical variance reduction literature (Greensmith et al., 2004; Peters & Schaal, 2008), accounts for heterogeneous gradient magnitudes but ignores cross-temporal correlations. With our Gradient-Norm Proxy:

$$\hat{B}_t^{\text{isolated}} = \frac{\sum_{i=1}^N G_t^{(i)} \cdot \hat{w}_t^{(i)}}{\sum_{i=1}^N \hat{w}_t^{(i)}}. \tag{52}$$

**Optimal Token Baseline (This Work).**    Our OTB relaxes both assumptions by incorporating cross-temporal correlations through the Gradient Consistency approximation:

$$\hat{B}_t^{\text{OTB}} = \frac{\sum_{i=1}^N G_t^{(i)} \cdot \hat{W}_t^{(i)}}{\sum_{i=1}^N \hat{W}_t^{(i)}}, \tag{53}$$

where $\hat{W}_t = \sum_{j=1}^t \hat{w}_j$ is the cumulative energy. The progression from value function to isolated energy to OTB reflects increasingly realistic modeling of gradient structure in deep networks.

*Table 2.* Hierarchy of baselines as relaxations of simplifying assumptions.

| Baseline | Cross-correlation | Score norm |
|---|---|---|
| Value function $V_t$ | $\langle s_k, s_t\rangle = 0$ | Constant |
| Isolated energy $B_t^{\text{isolated}}$ | $\langle s_k, s_t\rangle = 0$ | Varies |
| OTB $B_t^{\text{OTB}}$ | Gradient Consistency | Varies |

In Appendix F.5, we empirically compare OTB with the variant using isolated energy. The consistent performance gap (OTB > Isolated) validates that both cross-temporal correlations and heterogeneous gradient magnitudes are significant in Transformer-based LLM-RL, justifying our more complete modeling approach.

# D   Optimal Token Baseline for Off-Policy Policy Gradient

We extend the derivation of the OTB to the off-policy setting. Let $\pi_\beta$ denote the behavior policy that generated the trajectories, and $\pi_\theta$ denote the target policy we wish to optimize. The importance sampling ratio at step $t$ is defined as:

$$\rho_t = \frac{\pi_\theta(y_t \mid x, y_{<t})}{\pi_\beta(y_t \mid x, y_{<t})}. \tag{54}$$

To mitigate instability, we employ Truncated Importance Sampling (TIS) (Yao et al., 2025) with a clipped ratio $\bar{\rho}_t = \min(c, \rho_t)$. The off-policy estimator is defined as:

$$\tilde{g}_{\text{TIS}}(\tau) = \sum_{t=1}^{T} \bar{\rho}_t s_t (G_t - B_t). \tag{55}$$

However, the use of clipped weights $\bar{\rho}_t$ introduces bias, meaning $\mathbb{E}[\tilde{g}_{\text{TIS}}] \neq \nabla J(\theta)$. Therefore, minimizing variance alone is insufficient; we must minimize the **Mean Squared Error (MSE)**.

First, we establish the decomposition of the MSE by adding and subtracting the expectation $\mathbb{E}[\tilde{g}_{\text{TIS}}]$ inside the norm:

$$\begin{aligned}
\text{MSE}(\tilde{g}_{\text{TIS}}) &= \mathbb{E}\left[\|(\tilde{g}_{\text{TIS}} - \mathbb{E}[\tilde{g}_{\text{TIS}}]) + (\mathbb{E}[\tilde{g}_{\text{TIS}}] - \nabla J)\|^2\right] \\
&= \underbrace{\mathbb{E}[\|\tilde{g}_{\text{TIS}} - \mathbb{E}[\tilde{g}_{\text{TIS}}]\|^2]}_{\text{Variance}} + \underbrace{\|\mathbb{E}[\tilde{g}_{\text{TIS}}] - \nabla J\|^2}_{\text{Bias}^2} + \underbrace{2\mathbb{E}\left[\langle \tilde{g}_{\text{TIS}} - \mathbb{E}[\tilde{g}_{\text{TIS}}], \mathbb{E}[\tilde{g}_{\text{TIS}}] - \nabla J\rangle\right]}_{\text{Cross-Term}}.
\end{aligned} \tag{56}$$

The Cross-Term vanishes as the expected deviation from the mean, $\mathbb{E}[\tilde{g}_{\text{TIS}} - \mathbb{E}[\tilde{g}_{\text{TIS}}]]$, is zero. Thus, $\text{MSE} = \text{Variance} + \text{Bias}^2$.

We can then simplify the MSE as follows:

$$\begin{aligned}
\text{MSE}(\tilde{g}_{\text{TIS}}) &= \mathbb{E}[\|\tilde{g}_{\text{TIS}} - \mathbb{E}[\tilde{g}_{\text{TIS}}]\|^2] + \|\mathbb{E}[\tilde{g}_{\text{TIS}}] - \nabla J\|^2 \\
&= \mathbb{E}[\|\tilde{g}_{\text{TIS}}\|^2] - \|\mathbb{E}[\tilde{g}_{\text{TIS}}]\|^2 + \|\mathbb{E}[\tilde{g}_{\text{TIS}}]\|^2 - 2\langle \mathbb{E}[\tilde{g}_{\text{TIS}}], \nabla J\rangle + \|\nabla J\|^2 \\
&= \mathbb{E}[\|\tilde{g}_{\text{TIS}}\|^2] - 2\langle \mathbb{E}[\tilde{g}_{\text{TIS}}], \nabla J\rangle + \|\nabla J\|^2.
\end{aligned} \tag{57}$$

To find the optimal baseline, we differentiate with respect to $B_t$. While the third term $\|\nabla J\|^2$ is constant, the interaction term $-2\langle \mathbb{E}[\tilde{g}_{\text{TIS}}], \nabla J\rangle$ depends on the unknown true gradient. To proceed, we rely on the small bias approximation.

We observe that the baseline $B_t$ primarily acts as a variance reduction control variate, rather than a bias correction term.

- If the truncation is moderate, $\mathbb{E}[\tilde{g}_{\text{TIS}}] \approx \nabla J$, making the interaction term approximately constant $(-2\|\nabla J\|^2)$.

- Even with significant truncation, the sensitivity of the expected direction $\mathbb{E}[\tilde{g}_{\text{TIS}}]$ to changes in $B_t$ is negligible compared to the sensitivity of the variance $\mathbb{E}[\|\tilde{g}_{\text{TIS}}\|^2]$.

Under this assumption, $\nabla_{B_t}\langle \mathbb{E}[\tilde{g}_{\text{TIS}}], \nabla J\rangle \approx 0$. Consequently, minimizing the MSE is asymptotically equivalent to minimizing the Expected Squared Norm $\mathbb{E}[\|\tilde{g}_{\text{TIS}}\|^2]$.

Incorporating TIS into the policy gradient, we define the modified realized energy as:

$$W_t^{\text{TIS}} = \sum_{j=1}^{t} \bar{\rho}_t^2 w_t. \tag{58}$$

Base on the analysis in Appendix C (approximating cross-terms with realized energy), we minimize this norm yielding optimal baseline :

$$\min_{B_t} \mathbb{E}_{\tau \sim \pi_\beta}\left[(G_t - B_t)^2 W_t^{\text{TIS}}\right] \implies B_t^{\text{TIS}} = \frac{\mathbb{E}_{y_{\leq t} \sim \pi_\beta}[G_t \cdot W_t^{\text{TIS}}]}{\mathbb{E}_{y_{\leq t} \sim \pi_\beta}[W_t^{\text{TIS}}]}. \tag{59}$$

Finally, we apply our Logit-Gradient Proxy to approximate the modified realized energy as $\hat{W}_t^{\text{TIS}} = \sum_{j=1}^{t} \bar{\rho}_j^2 \hat{w}_j$, resulting in the practical off-policy Optimal Token Baseline:

$$\hat{B}_t^{\text{TIS}} = \frac{\sum_{i=1}^{N} G_t^{(i)} \cdot \hat{W}_t^{(i)\text{TIS}}}{\sum_{i=1}^{N} \hat{W}_t^{(i)\text{TIS}}}. \tag{60}$$

## E  Gradient Variance Proxy

All gradient variance analysis in this work relies on the standard decomposition of the variance for a stochastic gradient vector. For a stochastic gradient $\hat{g}$ with expectation $g_{\text{true}} = \mathbb{E}[\hat{g}]$, the total variance is defined as the expected squared deviation from the mean:

$$\mathbb{V}[\hat{g}] \triangleq \mathbb{E}\left[\|\hat{g} - \mathbb{E}[\hat{g}]\|^2\right]. \tag{61}$$

Expanding this quadratic form yields the well-known variance decomposition identity:

$$\mathbb{V}[\hat{g}] = \mathbb{E}\left[\|\hat{g}\|^2\right] - \|\mathbb{E}[\hat{g}]\|^2. \tag{62}$$

This identity decomposes the variance into the difference between the expected squared norm (total energy) and the squared norm of the expected gradient (signal energy).

In the context of policy optimization, the gradient estimator for a trajectory $\tau$ with a baseline $B$ is given by $\hat{g}(\tau) = \nabla_\theta \log \pi_\theta(\tau) \cdot A(\tau)$, where $A(\tau)$ is the approximated advantage. The squared Euclidean norm of this estimator is:

$$\|\hat{g}(\tau)\|^2 = \|\nabla_\theta \log \pi_\theta(\tau)\|^2 \cdot A(\tau)^2. \tag{63}$$

To evaluate this efficiently without performing a full backward pass for every sample, we employ our Logit-Gradient Proxy, denoted as $\hat{W}(\tau)$, to approximate the squared norm of the score function: $\hat{W}(\tau) \approx \|\nabla_\theta \log \pi_\theta(\tau)\|^2$. This yields a computationally efficient estimator for the gradient energy of a single trajectory:

$$\|\hat{g}(\tau)\|^2 \approx \hat{W}(\tau) \cdot A(\tau)^2. \tag{64}$$

Base on this, we decompose the statistics for a mini-batch of size $N$ into three components to monitor training dynamics:

- **Mean Squared Magnitude (Total Power, $P_{\text{total}}$):** This estimates the second moment of the gradient distribution, representing the total energy (Signal + Noise) averaged over the batch.

$$P_{\text{total}} = \frac{1}{N} \sum_{i=1}^{N} \hat{W}(\tau_i) \cdot A(\tau_i)^2. \tag{65}$$

- **Squared Norm of the Empirical Mean (Signal Strength, $S$):** This measures the magnitude of the empirical mean gradient vector. It serves as a proxy for the strength of the consensus update direction.

$$S = \left\|\frac{1}{N} \sum_{i=1}^{N} \hat{g}(\tau_i)\right\|^2. \tag{66}$$

- **Estimated Variance of the Mean ($\widehat{\mathbb{V}}[\bar{g}]$):** We estimate the variance of the batch mean gradient. High values indicate significant disagreement among trajectories, signaling potential training instability. Using the unbiased estimator for the variance of the mean derived from the batch statistics $P_{\text{total}}$ and $S$:

$$\widehat{\mathbb{V}}[\bar{g}] = \frac{1}{N-1}\left(P_{\text{total}} - S\right). \tag{67}$$

We utilize these proxies throughout our experiments to demonstrate that the Optimal Token Baseline minimizes $\widehat{\mathbb{V}}[\bar{g}]$, thereby improving training stability.

# F   Detailed Experimental Results

In this section, we present the detailed experimental settings and additional empirical results to demonstrate the superior training stability of the Optimal Token Baseline.

## F.1   Detailed Settings

We implement Optimal Token Baseline on top of the VeRL[1] framework. Our experiments cover two distinct paradigms: Single-Turn Reasoning (math) and Multi-Turn Tool-Integrated Reasoning (TIR), as shown in Figure 15. For TIR training, we follow the setup from SimpleTIR (Xue et al., 2025), requiring the LLM to generate Python code during the reasoning process and using Sandbox Fusion as an interpreter to execute the generated code. We utilize rule-based reward signal to optimize the policy for both paradigms.

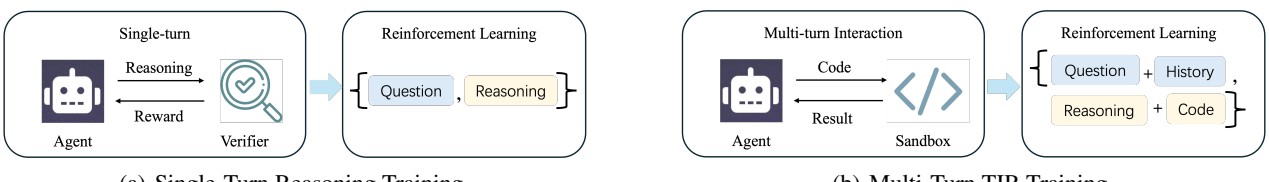

(a) Single-Turn Reasoning Training                     (b) Multi-Turn TIR Training

*Figure 15.* Overview of Single-Turn Reasoning and Multi-Turn TIR training in our experimental settings.

For training, we employ unaligned base models from the Qwen series and use the deduplicated DAPO-MATH-17k as the training dataset. We adopt a full on-policy RL paradigm, where the update batch size is equivalent to the rollout batch size. We fix all hyperparameters uniformly, including a batch size of 128, a maximum response sequence length of 8192, a group size of 16, and a learning rate of $10^{-6}$. Our training objective only optimizes the policy loss, with no KL divergence loss or entropy regularization term incorporated. During the rollout phase, both the sampling temperature and the Top-p are set to 1.0. For TIR settings, we set the maximum number of interaction turns to 5. These standardized training configurations are formalized in Table 3.

*Table 3.* Hyperparameter Training Settings

| Hyperparameter | Value |
|---|---|
| Learning Rate | $1 \times 10^{-6}$ |
| Optimizer | AdamW |
| Update Batch Size | 128 |
| Rollout Batch Size | 128 |
| Group Size ($N$) | 16 (4 for ablation) |
| Max Sequence Length | 8192 |
| KL Coefficient | 0.0 |
| Entropy Bonus | 0.0 |
| Rollout Temperature | 1.0 |
| Rollout Top-$p$ | 1.0 |
| Maximum Turns | 5 (for TIR) |

For evaluation, we conduct a comparison on a suite of widely-used benchmarks: MATH500, AMC23, AIME24, and AIME25. We set the generation temperature to 1.0 and the Top-p to 0.95, and report the avg@32 scores.

## F.2   Detailed Training Curves

We first establish the superior performance of the Optimal Token Baseline across all benchmarks, as presented in Table 1. To further substantiate the notable training stability of OTB—a key advantage over competing approaches—we visualize the full training dynamics of all methods in Figure 16. Specifically, we report the training curves for the AIME24 and AIME25, as we logged the real-time training performance of these two datasets.

---

[1]https://github.com/volcengine/verl

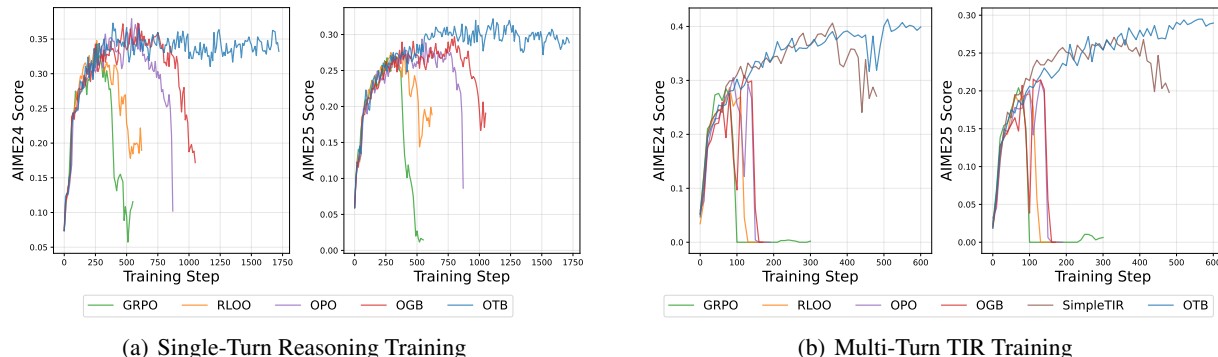

(a) Single-Turn Reasoning Training   (b) Multi-Turn TIR Training

*Figure 16.* Training curves for all methods.

As observed in the visualization, all comparative methods are susceptible to training collapse, with this phenomenon being exacerbated under TIR setting. With the exception of SimpleTIR, competing approaches exhibit premature training collapse within approximately 100 training steps. Even SimpleTIR, however, is unable to evade training collapse, which fundamentally prevents it from achieving higher scores. In contrast, OTB attains the highest performance across all evaluated benchmarks by maintaining stable training dynamics throughout the entire optimization process. This finding underscores that RL training stability is a critical prerequisite for unlocking the advanced reasoning potential of LLM.

### F.3   Training Stability of OTB

RL training instability stems from bias and variance in gradient estimation during policy updates. To quantitatively characterize training stability, we monitor a suite of critical metrics spanning both bias and variance perspectives.

For the bias metrics, we measure the Training-Rollout Mismatch between the training policy $\pi_\theta$ and the behavioral policy $\pi_\beta$. Given that our experiments are conducted on dense models with full on-policy updates, the mismatch can only be attributed to the engine discrepancies between the training (FSDP) and the rollout (vLLM). In addition to logging the KL divergence between the two policies, we also record the log absolute perplexity (PPL) gap as follows:

$$\frac{1}{N}\left|\frac{1}{T}\sum_{t=1}^{T}\log\pi_\theta(y_t\mid x,y_{<t}) - \frac{1}{T}\sum_{t=1}^{T}\log\pi_\beta(y_t\mid x,y_{<t})\right|, \tag{68}$$

where $N$ denotes the batch size and $T$ denotes the response length. For the variance metrics, we track two core proxies: the gradient variance calculated via the method detailed in Appendix E, and the gradient norm. Beyond these bias and variance metrics, we also observe additional key indicators for assessing overall training stability, including the average policy entropy and the response length.

As illustrated in Figures 17 and 18, the Optimal Token Baseline maintains consistent stability across all monitored metrics throughout the training process, enabling sustained optimization over extended training steps without experiencing catastrophic collapse. Notably, under the TIR settings—where response lengths exhibit substantial variability within a single group, introducing additional noise and optimization challenges—OTB demonstrates significantly superior stability compared to all competing methods. This empirical observation underscores that OTB's stability advantages are not only consistent across tasks but also amplified in high-variance training regimes, validating its robustness as a general-purpose baseline for LLM-RL.

### F.4   Results on Larger Model

To further demonstrate the superiority of the Optimal Token Baseline, we conduct extensive experiments on larger-size models, specifically employing Qwen3-8B-Base for Multi-Turn TIR and Qwen3-14B-Base for Single-Turn Reasoning. As illustrated in Figure 19, OTB consistently maintains stable training evidenced by the smooth gradient norm, even as model size expands and optimization complexity escalates. Critically, this sustained training stability directly translates to substantially higher scores relative to competing methods. These results underscore that OTB's stability generalize robustly to larger-scale architectures, validating its practical utility for LLM-RL.

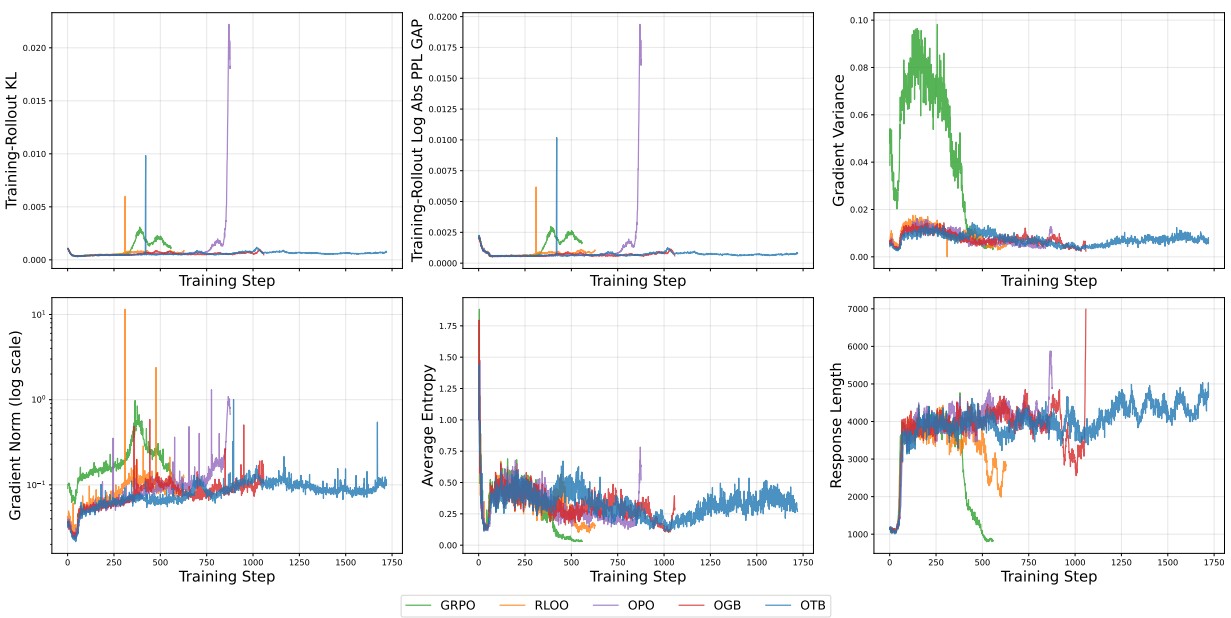

*Figure 17.* Training metrics for all methods under Single-Turn Reasoning.

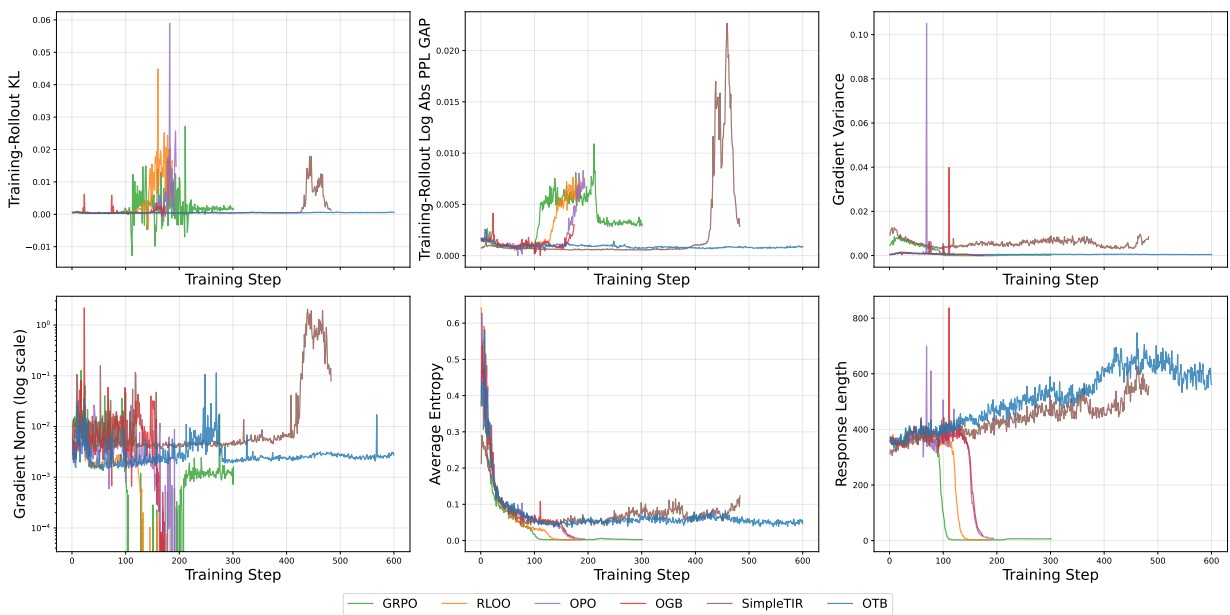

*Figure 18.* Training metrics for all methods under Multi-Turn TIR Training.

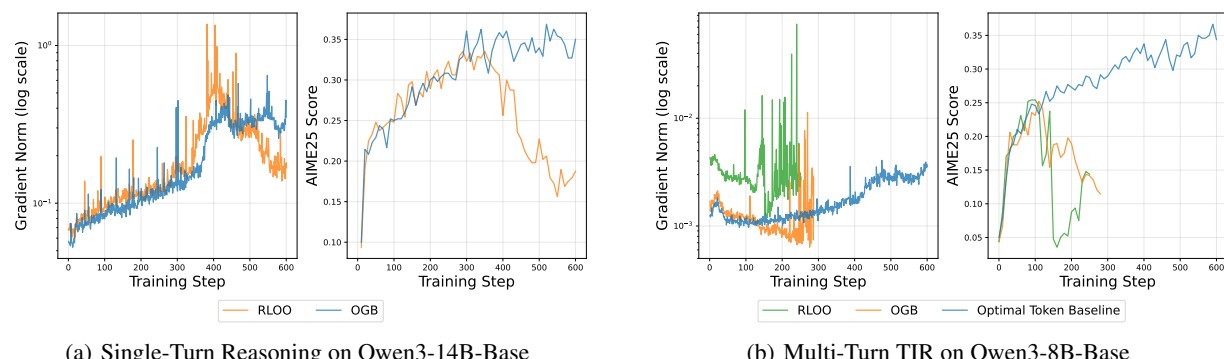

(a) Single-Turn Reasoning on Qwen3-14B-Base     (b) Multi-Turn TIR on Qwen3-8B-Base

*Figure 19.* Results under larger model.

## F.5   Comparison with OTB Variants

In Appendix C.7, we analyze a variant of the Optimal Token Baseline that employs *isolated energy* rather than realized energy as the weighting mechanism. This simplified variant relies on the assumption that the current score function $s_t$ is independent of past score functions $s_{k<t}$. Figure 20 presents an empirical comparison between this "OTB w. Isolated Energy" variant and our proposed OTB (which utilizes realized energy). As illustrated, the isolated energy variant consistently fails to match the performance of the proposed OTB across both the Single-Turn Reasoning and TIR settings. It suggests that the current score function is significantly correlated with past scores, thereby invalidating the independence assumption. Conversely, this evidence validates the gradient consistency assumption central to our derivation. It underscores that the efficacy of OTB stems from its ability to account for accumulated noise and historical dependencies via the realized energy, rather than treating each token's gradient contribution in isolation.

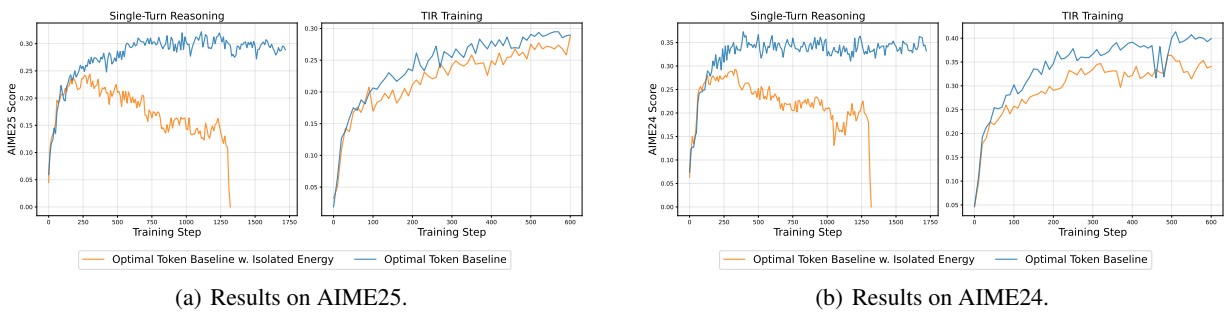

(a) Results on AIME25.     (b) Results on AIME24.

*Figure 20.* Comparison between the proposed Optimal Token Baseline and the variant using isolated energy.

