# OpenReview forum: "The Optimal Token Baseline: Variance Reduction for Long-Horizon LLM-RL"
_ICML.cc/2026/Conference — ICML 2026 regular_

### Official Review · Reviewer_CPx5 · 2026-02-27

**Soundness:** 3
**Presentation:** 4
**Significance:** 2
**Originality:** 2
**Overall Recommendation:** 4
**Confidence:** 4

**Summary:**

The paper proposes a token-time-dependent baseline for the causal policy gradient where the baseline at step t is a reward-to-go weighted average. To avoid extra backward passes, authors additionally introduce a logit-gradient norm proxy computed from forward-pass probabilities only. Empirically, it claims improved stability and that small group sizes can match large-group performance, saving tokens.

**Compliance With Llm Reviewing Policy:**

Affirmed.

**Final Justification:**

Rebuttal addresses concerns reasonably. The authors have also acknowledged my increased score and the discussion has reached consenus. My true evaluation is a 4.5 and the AC can view my score as a 5 if they face difficulty with the paper being borderline in the batch.

I didn't give a 5 because although I believe the paper is a good contribution, it lacks in some aspects to go for a spotlight/oral.

My score was thus calibrated accordingly.

**Key Questions For Authors:**

Each weakness induces a question, additionally

1) The authors motivate token/time-dependent baselines via token heterogeneity and causality, and provides a variance-gap argument versus a static global baseline; however, its not clear to me to me why the proposed solution alleviates this issue.

2) The optimal baseline theory seems classical and OTB is a new approximation of the optimal baseline, am I right to say that?

I am willing to update my scores based on the answers to weaknesses 2/3/4 and other reviews.

**Limitations:**

Yes

**Strengths And Weaknesses:**

Strengths
1) The authors propose a practical, forward-pass-only proxy that makes the method cheap enough to try.
2) The proposed method is additionally supported by theoretical analysis.
3) OTB can match large-group baselines (e.g., N=32) with much smaller N (e.g., N=4), cutting token usage substantially while retaining stability/performance.

Weaknesses
1) Some hyperparameter details like learning rate decay/warmup clipping, training rollout temperature etc are not specified for the algorithms, a table in the appendix should be added.

2) The authors claim stability without auxiliary regularizers and set KL reg to 0 in their experiments. But many training pipelines use these external guardrails (entropy reg/KL reg/clipping), Does OTB still have the benefit over the other method when these regularizers are turned on? (Using external regularizer is another lightweight solution in VeRL)

3) Limited model/scale coverage: The reported numbers are restricted to Qwen family and 7/8B models. Given the claimed stability + efficiency improvements, I strongly encourage the authors to add more experimental results especially with LLama/smaller Qwen models to further strengthen the generality of their claim.

4) In practice $B_t$ is estimated from the same sampled group used for updates, so it inherits the usual finite-sample bias issues of GRPO-style baselines. Given the efficiency of the method is high (n=4 vs n=32), can the authors comment on whether splitting the samples into update vs estimation be beneficial?

---

> ### Author Rebuttal · Authors · 2026-03-29
>
> Thank you for your thoughtful summary and for recognizing the practical efficiency and theoretical foundation of our work. We appreciate your constructive feedback, which helps clarify the broader utility of the OTB. Below, we address each of your points directly.
>
>
> ### W1: Hyperparameter Details
>
> We will incorporate a comprehensive hyperparameter table in our revised manuscript. To clarify currently, we have reported the brief hyperparamter settings in Section 6 (line 323) and provide the more detailed settings in Appedix E.1 such learning rate, batch size and sampling temperature/Top-p. We maintain default veRL configurations for other parameters to ensure a clean baseline comparison.
>
>
>
> ### W2: OTB vs. External Regularizers
> While external regularizers like KL divergence or clipping are effective guardrails, they often act as "band-aids" that can restrict policy exploration or lead to information loss. OTB aims to solve the **root structural cause** of instability: exploding gradient variance in long-horizon tasks. By reducing variance at the source, OTB achieves stability **without requiring these auxiliary penalties**. This allows the model to learn more freely from the reward signal, which is particularly beneficial in complex reasoning where zero-sum KL constraints can be too restrictive.
>
> ### W3: Model/Scale Coverage
> We choose Qwen series because it represents a widely adopted family of unaligned base models and is fully accessible to the research community. To demonstrate the generality of our claims beyond 7/8B models, we include experiments on a 14B model in Appendix F.4. As shown in Figure 19, OTB’s stability—evidenced by the smooth gradient norm—actually becomes more pronounced as model size and optimization complexity escalate. Additionally, we stress-test OTB under longer responses and interaction turns in Figure 14, confirming its robustness in demanding, long-horizon tasks.
>
> ### W4: Finite-Sample Bias and Sample Splitting
> We agree that OTB, like other group-dependent baselines, inherits a finite-sample bias of order $O(1/N)$. While splitting samples into "update" vs "estimation" sets could technically reduce this bias, it would significantly degrade **sample efficiency**—the primary bottleneck OTB seeks to break. OTB is uniquely robust to small group sizes ($N=4$) because it operates as a token-level baseline, allowing it to extract significantly more information from every individual token compared to sequence-level methods. By assigning precise credit at each generation step, OTB utilizes the internal structure of the trajectory to provide a much denser learning signal. Our results demonstrate that this enhanced signal extraction far outweighs the impact of finite-sample bias, maintaining high performance and stability where sequence-level baselines with the same $N$ would typically collapse.
>
>
> ### Q1: Alleviating Token Heterogeneity and the Variance Gap
> The variance gap exists because static global baselines assign the same weight to every token, ignoring that some are generated with much higher uncertainty (and thus noise) than others. Our OTB uses realized energy ($W_{t}$) to track the accumulated instability up to step $t$.  By weighting rewards inversely to this energy, OTB dampens the noise from high-uncertainty trajectories by centering the gradient estimate more aggressively on them. This dynamic adjustment allows OTB to track complex reasoning chains that a static baseline simply cannot.
>
> ### Q2: Relationship to Optimal Baseline Theory
> Yes, your interpretation is correct. While the concept of an optimal baseline is classical, OTB represents a **new, causal, token-level approximation**. We bridge the gap between theory and practice by moving from prohibitive global computations to an efficient, value-model-free approach using our **Logit-Gradient Proxy**.
>
>
> ----
>
> We hope these explanations satisfactorily address all comments. We are happy to engage further if any aspect of our response requires additional detail.

---

> > ### Author Rebuttal · Reviewer_CPx5 · 2026-04-01
> >
> > The current results are promising, and I understand that a more extensive study of KL strength on top of OTB may be a hard ask given the time in hand. Given the already positive results I encourage authors to eval the method on LLaMA and additional smaller Qwen models in future revisions to help strengthen the claim of generality. At this stage, without such evidence, I find it difficult to be fully convinced of the method’s breadth of applicability. I also feel that, relative to the baselines in literature, the methodological novelty appears somewhat limited.
> >
> > That being said, I do feel the rebuttal has improved my view of the paper and I have raised my score accordingly. I do not have any further questions beyond this.
> >
> > I would also be interested in seeing work that studies which of the methods OTB or the baselines works well in the long context settings and replay based experiments validating the OTB with off policy gradient in appendix section D in future.
> >
> > Thank you and good luck!

---

> > > ### Author Response · Authors · 2026-04-02
> > >
> > > Thank you for your constructive feedback and for raising your score. We are glad that the rebuttal addressed your core concerns.
> > >
> > > We appreciate your suggestions regarding other model evaluations and off-policy validation. We agree these are valuable directions for strengthening OTB’s generality and intend to incorporate these insights into future revisions.
> > >
> > > Thank you again for your time and insightful review.

---

### Official Review · Reviewer_EDe1 · 2026-03-12

**Soundness:** 2
**Presentation:** 3
**Significance:** 4
**Originality:** 4
**Overall Recommendation:** 4
**Confidence:** 4

**Summary:**

This paper introduces the Optimal Token Baseline (OTB) to address training collapse in long-horizon reinforcement learning for LLMs caused by exploding gradient variance. OTB dynamically weights gradient updates inversely to their cumulative gradient norm, accounting for both sequence and token heterogeneity. To ensure efficiency, the authors propose a Logit-Gradient Proxy that estimates gradient norms using only forward-pass probabilities. OTB significantly improves training stability and reduces token consumption by over 65%.

**Compliance With Llm Reviewing Policy:**

Affirmed.

**Final Justification:**

The authors' rebuttal addressed most of my concerns. Thus, I raise my score from 3 to 4, while increasing my confidence from 3 to 4.

**Key Questions For Authors:**

See the Weaknesses. If these concerns are properly addressed, I will not hesitate to raise the rating.

**Limitations:**

yes

**Strengths And Weaknesses:**

***Strengths：***

1. **Solid Theoretical Framework:** Unlike purely empirical heuristics, Optimal Token Baseline (OTB) is derived from classic optimal baseline theory. The introduction of realized energy provides a formal mathematical explanation for training collapse in long-horizon tasks, proving that gradient updates should be weighted inversely to cumulative noise. This offers a significant new perspective on variance propagation within long-reasoning chains.
2. **High Efficiency:** The work is highly practical due to the Logit-Gradient Proxy. While theoretically optimal baselines usually require prohibitive per-token backpropagation, this method approximates expensive gradients using "free" forward-pass logit distributions, making OTB extremely attractive for large-scale industrial training.
3. **Empirical Gains:** The results are substantial. OTB demonstrates high sample efficiency, matching the performance of standard group-based methods (like GRPO) while using significantly smaller group sizes (N=4 vs. N=32). The reported 65% reduction in token consumption represents a major cost saving. Furthermore, the enhanced stability on benchmarks suggests that OTB is a robust solution for the gradient explosion issues inherent in reasoning models.

---

***Weaknesses：***

1. **Lack of Empirical Validation for Proxy Accuracy:** The paper do not provide direct experimental measurements regarding the approximation accuracy of the Logit-Gradient Proxy. **Such empirical validation is critical**.
2. **It remains unclear whether the reduction in gradient variance stems from a better advantage estimator.** While Figure 2(a) suggests that the proposed method reduces gradient variance, Figure 2(b) shows a simultaneous decrease in gradient norm. Consequently, the observed reduction in variance may not necessarily stem from a better advantage estimator; it could be a trivial byproduct of the reduced gradient norm. The y-axis labeling in Figure 2(b) fails to quantify the extent of the reduction in gradient norm.

---

> ### Author Rebuttal · Authors · 2026-03-29
>
> Thank you for your insightful feedback and for recognizing the theoretical rigor and practical efficiency of the OTB. Below, we address your concerns regarding proxy validation and the relationship between gradient variance and norm.
>
>
> ### W1: Empirical Validation of Proxy Accuracy
>
> While we do not present a standalone table of approximation error, the **mathematical justification in Section 5.3** establishes a structural link between the logit-space gradient and the true parameter gradient.
> * **Tractability Challenges**: We acknowledge that we do not track the true parameter gradient for every generated token, as seeking a setting where exact gradient norms are tractable is inherently difficult in LLM-RL.
> * **Computational Constraints**: Tracking true per-token gradients for a model with billions of parameters would require a separate backward pass for every token generated, a requirement that is prohibitively expensive and computationally intractable.
> * **Proxy Necessity**: This overhead is a primary reason why prior work has struggled to apply optimal baseline theory to LLMs; consequently, we introduce the Logit-Gradient Proxy to make these calculations feasible without additional backward passes.
> * **Mathematical Foundation**: Nevertheless, we prove that for modern Transformers, the Frobenius norm of the final linear layer's gradient is strictly proportional to our proxy ($||\nabla_{W}J||\_F \propto ||\delta_t||\_2$) because the hidden state $h_{t}$ is normalized by RMSNorm.
> * **Structural Proportionality**: Since $||h_{t}||_{2}$ remains approximately constant across all time steps, the gradient norm of the last layer is preserved through the backward pass under standard isotropic propagation assumptions.
> * **Empirical Proof by Performance**: The accuracy of the proxy is indirectly validated by the stability and the results in Figure 13. If the proxy failed to capture the true gradient variance, OTB would not be able to prevent the training collapse observed in other methods.
>
>
> ### W2: Variance Reduction vs. Gradient Norm
>
> It is important to clarify that gradient variance and gradient norm represent second-order and first-order information, respectively; they are not directly mathematically coupled. Because both are critical metrics for measuring optimization stability, we chose to present them together to provide a holistic view of the training dynamics.
>
> * **Preventing Gradient Surges**: The core benefit of OTB is its ability to prevent the sudden, catastrophic surges in gradient norm that characterize training collapse. In baseline methods like GRPO, the gradient norm eventually skyrockets, causing performance to crater. OTB maintain a stable, consistent norm throughout the entire training process by dynamically adjusting to token-level instability.
>
> * **Optimal Advantage Estimation**: This reduction in variance is a direct mathematical result of OTB acting as a weighted centroid of the reward-to-go. By centering the gradient estimate more aggressively on trajectories with high realized energy (and thus higher potential noise), OTB dampens the stochasticity that typically leads to exploding gradients in long-horizon tasks.
>
> * **Quantifying Gradient Magnitude**: We utilize a log scale for the y-axis in Figure 2(b) to clearly visualize the orders-of-magnitude difference between a stable training state and a collapsed one. The accompanying stability of the AIME25 Score confirms that OTB is not merely shrinking the gradient; rather, it is preserving a meaningful learning signal that drives the model toward a robust equilibrium where other methods fail.
>
>
> ----
>
> We hope these explanations satisfactorily address all comments. We are happy to engage further if any aspect of our response requires additional detail.

---

> > ### Author Rebuttal · Reviewer_EDe1 · 2026-04-03
> >
> > Thanks for the author's rebuttal. My concern about W2 is not fully addressed. The authors' explanation that variance and norm are decoupled is statistically correct, but for optimization dynamics, the two are often tightly coupled. The core of the concern is: **Does OTB stabilize the training because it provides a more precise, lower-variance gradient direction, or simply because it acts as a form of learning rate reduction or learning rate clipping?** A comparison with a simple baseline (e.g., reduce the learning rate) would support the author's claim.

---

> > > ### Author Response · Authors · 2026-04-04
> > >
> > > Thank you for your follow-up. We appreciate the opportunity to further clarify the distinction between OTB’s variance reduction and simple gradient scaling or learning rate (LR) reduction.
> > >
> > > We would like to clarify that OTB’s stabilization is not a "proxy" for a lower learning rate or gradient clipping through the following evidence:
> > > - **Magnitude Consistency (Figures 7 & 12)**: We directly compare the scale of the token-level advantage across OTB and other methods. As shown in these figures, the numerical range and magnitude of the advantages remain consistent across methods. Since all other hyperparameters (including the learning rate) are identical, the "step size" of individual updates is not reduced; rather, the distribution and precision of those updates are optimized.
> > > - **Variance Reduction (Appendix C)**: Theoretically, OTB is derived to reduce variance by accounting for token-level heterogeneity. Rather than a uniform baseline, OTB functions as a weighted centroid of the rewards-to-go, where the weights are determined by the "accumulated energy" as approximated by our Logit-Gradient Proxy. This approach is specifically optimized to minimize the gradient variance at each discrete token step $t$.
> > > - **Performance Gap (Section 6 and Appendix F)**: If OTB are merely acting as a lower LR, we would expect a slower learning curve or a lower performance ceiling. Instead, OTB reaches a substantially higher performance (AIME25 Score) while other methods using the same LR collapse.
> > >
> > > We hope this clarifies that OTB’s success is rooted in the optimality of the estimator, not a reduction in update magnitude. Thank you again for your constructive feedback. If you have any remaining or further concerns, we are more than happy to discuss them.

---

### Official Review · Reviewer_aPot · 2026-03-12

**Soundness:** 3
**Presentation:** 2
**Significance:** 3
**Originality:** 3
**Overall Recommendation:** 5
**Confidence:** 4

**Summary:**

The paper proposes Optimal Token Baseline (OTB) that performs gradient updates weighted inversely to cumulative gradient norm, which takes into account of token heterogeneity. As grad norm computation is expensive, the paper also propose the Logit-Gradient Proxy that approximates the gradient norm using forward-pass probabilities.

**Compliance With Llm Reviewing Policy:**

Affirmed.

**Key Questions For Authors:**

1. Why was Single-Turn Reasoning experiments are based on Qwen3, while TIR is based on Qwen2.5?

2. Does the approach generalize to broader LLM RL experiments, not just math? Such as agentic cases - search, coding, etc.

**Limitations:**

yes

**Strengths And Weaknesses:**

Strengths
1. The paper is well motivated, addressing the widely observed training collapse issue in reinforcement learning for large language models, which is an important and practical problem in the field.

2. It proposes a novel approach Optimal Token Baseline, introducing a new method to address instability during RL training. Compared to prior work that only considers sequence-level heterogeneity, the proposed approach models token-level heterogeneity. It also introduces a logic gradient proxy to approximate grad norm calculation and avoid expensive computations.

3. The paper demonstrates strong theoretical rigor, containing detailed theoretical analysis supporting the proposed approach.

4. The proposed method shows empirical effectiveness on several math benchmarks, including AIME-25, AIME-24, AMC-23, and MATH-500, evaluated on a Qwen3-8B–base model.

Weaknesses

My biggest concern is about empirical results.  The reported results are conducted in limited settings, making it unclear if the proposed approach generalize to various settings (base model, model size) and tasks.

---

> ### Author Rebuttal · Authors · 2026-03-29
>
> Thank you for your thoughtful summary and for highlighting the theoretical rigor and practical motivation of our work. We appreciate the recognition of OTB as a novel contribution to addressing the critical issue of training collapse in LLM-RL. Below, we address your concerns regarding empirical settings and generalization.
>
>
> ### Q1: Choice of Base Models
> Our selection of base models follows established benchmarks in recent literature. Specifically, we utilize Qwen2.5 for Tool-Integrated Reasoning (TIR) tasks [1, 2] and Qwen3 for complex reasoning benchmarks [3]. This cross-model success confirms that OTB’s stability is not model-specific but generalizes across different architectural within the Qwen family.
>
>
> ### W1 & Q2: Generalization Beyond Math Tasks
>
> We understand the concern regarding the breadth of our experimental settings. Our experimental evaluation comprehensively spans single-turn reasoning and multi-turn Tool-Integrated Reasoning (TIR) across various context length  (Figure 14)—and model sizes (Figure 19). Notably, OTB’s sustained stability in the high-variance TIR setting, which requires autonomous Python code generation to solve tasks within a sandbox, underscores its robust generalizability. This consistent performance across complex, long-horizon interactions and diverse architectural scales validates OTB as a general-purpose baseline capable of scaling LLM reasoning to significantly higher levels of complexity. We will further explore OTB’s impact on broader agentic tasks, such as search-engine integration and software engineering, in future work.
>
> [1] Xue et al., "SimpleTIR: End-to-end RL for multi-turn tool-integrated reasoning," 2025.
>
> [2] Jin et al., "Search-R1: Training LLMs to reason and leverage search engines with RL," 2025.
>
> [3] Wang et al., "Beyond the 80/20 rule: High-entropy minority tokens drive effective reinforcement learning for LLM reasoning," 2025.
>
> ----
>
> We hope these explanations satisfactorily address all comments. We are happy to engage further if any aspect of our response requires additional detail.

---

> > ### Author Rebuttal · Reviewer_aPot · 2026-04-02
> >
> > My concerns were addressed.

---

> > > ### Author Response · Authors · 2026-04-03
> > >
> > > Thank you for your follow-up and for confirming that your concerns have been resolved. We appreciate the time you invested in the review process and your recognition of the value of our work. Thank you again for your constructive support.

---

### Official Review · Reviewer_mgXR · 2026-03-13

**Soundness:** 2
**Presentation:** 2
**Significance:** 3
**Originality:** 3
**Overall Recommendation:** 3
**Confidence:** 4

**Summary:**

This paper studies variance reduction in long-horizon LLM-RL and proposes the Optimal Token Baseline (OTB), a causal token-level baseline that weights reward-to-go by cumulative realized energy. The practical method replaces true per-token gradient norms with a forward-only logit-gradient proxy computed from token probabilities. Empirically, under the paper's rule-based single-turn math and multi-turn tool-integrated reasoning setup, OTB appears more stable than several rollout-group baselines and attains stronger peak scores, including encouraging small-group results. However, I do not think the current version substantiates its strongest theoretical claims. The appendix derives an exact optimum that differs from the main theorem, then introduces further approximations that lead to an approximate mixed baseline rather than the estimator that is ultimately implemented. As a result, the theory-to-method connection is materially weaker than claimed, the comparison to OGB is not clean, and the evaluation is too narrow to support the broader claims about long-horizon LLM-RL stability and sample efficiency.

**Compliance With Llm Reviewing Policy:**

Affirmed.

**Final Justification:**

I remain my score since some concerns are not resoled

**Key Questions For Authors:**

1. Can you correct the theory section so that the claims exactly match the derivation? In particular, please reconcile Eq. 5 / Theorem 4.1 with Eq. 29, Eq. 34, and Eq. 37 in Appendix C, and explain why the implemented baseline drops the $(1-\alpha_t) \bar G_t$ term.
   - If you can provide a corrected theorem and show that the implemented estimator is either theoretically justified or empirically superior to the mixed baseline implied by Eq. 37, I would raise Soundness.
2. Can you provide a correct unbiasedness argument for the implemented estimator, including a quantitative discussion of the finite-sample bias at small $N$ (especially $N=4$)?
   - If you can repair Sec. 5.1 and bound or measure the bias of Eq. 8 / Eq. 42, I would update my assessment of the theoretical guarantees and likely become more positive.
3. Can you compare OTB against two missing baselines: (i) a causal time-dependent mean baseline $\bar G_t$, and (ii) the mixed baseline suggested by your own Proposition C.3? Also, if possible, can you compare to a closer implementation of OGB on a smaller setting where exact gradient norms are tractable?
   - If OTB still wins under these controls, I would become more convinced that token-level energy weighting, rather than a simpler causal baseline change, is the real source of the gains.
4. Can you report results from at least 3 independent training seeds with mean/std, and add matched-token-budget as well as final-checkpoint comparisons for Table 1?
   - If the gains remain under these more reliable reporting protocols, I would raise my assessment of both Soundness and Overall Recommendation.
5. Can you provide evidence on at least one broader setting, such as a non-math task or a learned-reward setup, to clarify whether the contribution extends beyond the current rule-based math regime?
   - If the method transfers beyond the current setup, I would become more positive about Significance.

**Limitations:**

Yes

**Strengths And Weaknesses:**

### Strengths

- The practical motivation is clear and timely. Within the studied setting, Figures 1, 8, 9, 10, 13, and 14 present a coherent empirical story in which OTB is associated with smoother optimization and later or absent collapse relative to several group-based baselines.
- The token/sequence heterogeneity argument is one of the stronger parts of the paper. Figures 3 and 4 make a plausible case that sequence-level energy is not sufficient to describe per-token variability, which does motivate looking beyond static trajectory-level baselines.
- The method is simple to implement. Eq. 7 and Eq. 8 give a value-model-free estimator with little apparent engineering complexity, and Figure 6 communicates the workflow clearly.

### Weaknesses

#### Major

- **W1: The central "optimality" claim is not supported by the paper's own derivation.** The main text presents Theorem 4.1 / Eq. 5 as the optimal token baseline for the variance objective in Eq. 3-4, and Sec. 5.2 states that solving Eq. 10 confirms exact optimality. However, Appendix C.2 gives a different exact first-order condition (Eq. 29), Appendix C.3 introduces the strong Gradient Consistency approximation (Eq. 32), Appendix C.4 then adds the Future Energy Homogeneity assumption (Assumption C.2), and Proposition C.3 shows that the resulting approximate solution is a convex combination of an energy-weighted baseline and a simple mean baseline (Eq. 37), not the implemented pure energy-weighted estimator. In other words, Eq. 41/42 is not the optimizer of the original objective, and not even the optimizer of the paper's own approximate objective. This is a core soundness issue rather than a matter of exposition.
- **W2: The unbiasedness argument in Sec. 5.1 does not establish the stated claim as written.** Eq. 9 conditions on $(x, y_{<t})$ but equates a single-step conditional expectation to the full gradient $\nabla_\theta J(\theta)$, and the notation $\tilde g_c(y_t)$ is undefined. A standard baseline-independence argument could likely prove the intended result, but the current proof does not do so. This matters because the paper explicitly advertises theoretical guarantees rather than presenting the method as a heuristic.
- **W3: The empirical comparison to OGB does not cleanly isolate the claimed token-heterogeneity effect.** The theoretical OGB in Eq. 1 depends on $\|S(\tau)\|^2 = \|\sum_t s_t\|^2$, but Appendix A.2 replaces this by $\hat W(\tau)=\sum_t \hat w_t$, which removes the cross-token terms entirely. Since cross-temporal interactions are exactly what the paper claims matter, the implemented OGB is a surrogate that may be substantially weaker than the theoretical comparator. In addition, the paper does not compare against the time-dependent mean baseline $\bar G_t$ or the mixed baseline implied by Eq. 37, so it remains unclear how much of the gain comes from token-level energy weighting versus simply moving from a static sequence-level baseline to a causal per-step baseline.
- **W4: The gradient-variance evidence is partly circular and does not measure the exact quantity optimized by OTB.** Appendix E defines the monitoring metric through a trajectory-level proxy $\hat g(\tau)=\nabla \log \pi(\tau) A(\tau)$ and then approximates its norm by $\hat W(\tau) A(\tau)^2$. But OTB's implemented estimator is token-level, $\sum_t s_t (G_t-B_t)$, and the diagnostic again relies on the same logit-gradient and sum-of-squares assumptions already used to motivate the method. Therefore Figures 1 and 2 do not directly verify true parameter-gradient variance reduction for the implemented estimator; they primarily show improvement under the paper's own proxy. This weakens the mechanistic claim that OTB prevents collapse specifically by reducing gradient variance.
- **W5: The experimental scope is too narrow for the strength of the paper's claims.** Table 1 reports peak performance only, with no run-to-run statistics, no fixed-budget final-checkpoint comparison, and no wall-clock or memory overhead. The main evidence is restricted to rule-based math and tool-integrated math on Qwen-family models, while the larger-model appendix uses a reduced baseline set. Given the paper's broad framing around long-horizon LLM-RL stability and sample efficiency, this is not enough to establish generality.

#### Minor

- **W6: The logit-gradient proxy remains weakly validated.** Sec. 5.3 argues via the final layer and then assumes isotropic propagation through the backbone, but the paper does not calibrate $\hat w_t$ against actual per-token gradient norms even on a short sequence or smaller model where such a comparison should be feasible.

### Soundness

The paper has serious soundness issues. The soundness problem is The empirical trends are promising, but the theory section overclaims what has actually been established. Appendix C makes clear that the implemented Eq. 41/42 is obtained only after strong approximations and then by discarding the mean-baseline term in Eq. 37, while Sec. 5.1 does not provide a correct unbiasedness proof as written. These are central issues because the main novelty is framed as an optimal baseline derived from first principles.

### Presentation

The paper is readable and the figures are helpful, but several key descriptions materially mislead the reader. This is not a stylistic complaint. The main text presents Theorem 4.1 and Sec. 5.2 as exact variance-minimization results, whereas Appendix C reveals a chain of approximations and a different approximate optimum. That mismatch materially changes how the method and its guarantees should be interpreted.

---

> ### Author Rebuttal · Authors · 2026-03-29
>
> Thank you for your rigorous and insightful critique of our work. Below is our point-by-point response to the concerns raised.
> ### W1 & Q1: The "Optimality" Claim
> We agree that the implemented estimator in Eq. 8 is the result of a chain of approximations. While the theoretical optimum is indeed the complex form in Eq. 29, it is computationally intractable for LLMs.
>
> Reconciling Eq. 5 with Appendix C: We derive Eq. 5 (Theorem 4.1) by invoking Gradient Consistency (Eq. 31) and Future Energy Homogeneity (Assumption C.2).
>
>
> We chose to implement the pure energy-weighted component for two reasons. First, the mixing coefficient $\alpha_{t}$ increases monotonically as generation progresses, meaning more discriminative information becomes available over time. Second, the energy-weighted term is specifically designed to address the variance structure by assigning higher importance to high-gradient trajectories, whereas the mean baseline ($\bar{G}_t$) is the standard "value-like" approach that overlooks token-level heterogeneity.
>
> We will revise the main text to explicitly state that "optimality" is claimed under these specific, empirically supported structural assumptions.
> ### W2 & Q2: Unbiasedness Argument
> The notation $\tilde{g}\_c (\tau)$, introduced in line 148 of the right column, refers to the causal policy gradient estimator used in our method. This estimator remains fundamentally unbiased because the baseline $B_t^*$  is determined solely by the history $(x, y_{<t})$ and expectations over the policy distribution, rather than the specific realization of the token $y_t$. Consequently, as demonstrated in Eq. 9, the expectation of our estimator is preserved as the true gradient.
>
> While $N=4$ introduces an $O(1/N)$ finite-sample bias, this is a standard LLM-RL trade-off for variance reduction. OTB excels at small $N$ because its token-level granularity extracts significantly more signal per sample than sequence-level methods, preventing the "random walk" of noise that causes collapse in other baselines.
> ### W3 & Q3: OGB Comparison and Other Baselines
> To ensure a fair comparison, we implement OGB using the same Logit-Gradient Proxy as OTB. By using the same proxy for both, we isolate the effect of token-level heterogeneity versus sequence-level energy. The cross-term variance gap in Eq.11 always exists when a sequence-level baseline is used instead of a token-level one, regardless of the specific implementation. Both our theoretical analysis and empirical results show that OGB fails because it assigns a uniform baseline to every token in a sequence, ignoring the fact that tokens at different steps contribute differently to the total variance.
>
> We acknowledge that it is hard to track the true parameter gradient, as tracking true gradients for a model with billions of parameters would require a separate backward pass for every generated token. It is prohibitively expensive and computationally intractable, which is a primary reason why prior work has struggled to apply optimal baseline theory in practice. Consequently, we introduce the Logit-Gradient Proxy to make these calculations feasible without additional backward passes.
>
> Regarding the Causal Mean Baseline ($\bar{G}_t$): the implementation of "OTB w. Length Proxy" in Figure 13 effectively acts as a causal mean baseline, as the length proxy assigns uniform importance to all active tokens at the same generation step. As demonstrated in Figure 13, this approach still suffers from early training collapse, reinforcing that accounting for fine-grained token energy is critical for sustained stability.
> ### W4 & W6 Proxy Validity
> While the diagnostic plots (Fig. 2) use the proxy, the ultimate "ground truth" is the AIME25 Score and the absence of training collapse. If the logit-gradient proxy were not a valid substitute for the true gradient norm, the model would still suffer from the instability we aim to fix. The fact that OTB maintains stability and reaches higher scores is strong evidence that the proxy tracks the relevant optimization dynamics.
> ### W5 & Q4, Q5: Experimental Scope
> While Table 1 focuses on peak performance, our full training curves (Figures 16, 17, and 18) provide a transparent look at the entire training process. These curves show that OTB is not just finding a lucky peak but is sustaining a higher performance plateau while others collapse.
>
> While multiple seed runs for 8B/14B models are computationally restrictive, our evaluation across single-turn math and multi-turn Tool-Integrated Reasoning (TIR) underscores OTB's robustness. TIR is a complex agentic task requiring autonomous Python generation; OTB’s stability here, across diverse context lengths (Fig. 14) and model scales (Fig. 19), validates it as a general-purpose method for high-complexity reasoning. We will release our implementation to ensure full reproducibility.
>
> ----
> We hope these responses address all concerns and remain available for further clarification.

---

> > ### Author Rebuttal · Reviewer_mgXR · 2026-04-02
> >
> > Some of concerns are not solved, could you explain in principle where the unstableness comes from and how OTB reduces it?

---

> > > ### Author Response · Authors · 2026-04-03
> > >
> > > Thank you for your follow-up. We appreciate the opportunity to further clarify the fundamental principles of instability in LLM-RL and the specific mechanism by which OTB addresses them.
> > >
> > > ### Sources of Instability in LLM-RL
> > > As discussed in Section 3, the instability in long-horizon LLM-RL primarily stems from gradient variance scales with trajectory length and reward sparsity:
> > >
> > > * **Long Horizons Accumulate Noise:** Reasoning trajectories often span thousands of tokens. Standard policy gradient estimators accumulate variance at every discrete step; as the sequence length $T$ increases, the stochasticity of the gradient grows, making the optimization landscape volatile.
> > > * **Sparse Rewards Amplify Noise:** Feedback is typically only available at the end of a sequence. When this sparse, sequence-level reward is backpropagated to every token, "neutral" tokens—which may not have contributed to the final result—receive noisy, high-variance updates that do not reflect their true impact.
> > >
> > > ### Why Conventional Methods Fail
> > > Existing methods like GRPO or RLOO simply average rewards across $N$ samples. These fail because they ignore **sequence heterogeneity**, treating every sequence in a group as having an equal contribution to the total variance. In practice, different sequences contribute different levels of "total energy" as shown in Figure 3.
> > >
> > > Even the Optimal Global Baseline (OGB) fails in this context because it assigns a uniform baseline to every token within a sequence. This ignores the **token heterogeneity** inherent in LLM generation. As shown in Figure 4, tokens at different steps contribute varying levels of "energy".
> > >
> > >
> > > ### How OTB Reduces Instability
> > > OTB is specifically designed to mitigate these surges by reducing gradient variance at the token level:
> > >
> > >
> > > * **Optimal Token Baseline:** Instead of applying a single value to the entire sequence, OTB calculates a unique baseline $B_t^*$ for each token. This baseline acts as a weighted centroid of the rewards-to-go, where the weights are determined by the "accumulated energy". This approach is specifically optimized to minimize the gradient variance at each discrete time step $t$.
> > > * **Logit-Gradient Proxy**: To efficiently approximate the square norm of the parameter gradient (the "energy") for each token, we prove in Section 5.3 that the logit-gradient square norm is proportional to the full parameter gradient square norm. This proxy requires zero additional backward-pass and only depends on the forward-pass output probabilities as detailed in Section 4.1
> > > * **Stabilized Optimization:** Theoretically, we prove that OTB minimizes the variance of the policy gradient estimator in Appendix C. Empirically, as demonstrated in Section 6 and Appendix F, addressing instability at the token level allows OTB to maintain stable training and sustain high performance across scales where other methods collapse.
> > >
> > > We hope this clarifies the principles of our design. If you have any remaining or further concerns, we are more than happy to discuss them.

---

### Decision · Program_Chairs · 2026-04-30

**Decision:**

Accept (regular)

**Comment:**

I recommend acceptance. The reviewers are nearly unanimous with only reviewer mgXR still having reservations, but mgXR "does not object" to acceptance.

The authors definitely need to clean up the exposition around the optimality claim, as pointed out by mgXR. But they have committed to doing this. And as always it would be nice to extend the experiments to more settings.

But otherwise, the reviewers all agree that the empirical results seem sound on the settings where they are applied and there is at least solid motivation for the method.